# Holocene climate change in southern Oman deciphered by speleothem records and climate model simulations

Ye Tian [1], Dominik Fleitmann [2], Qiong Zhang [3], Lijuan Sha[1], Jasper. A. Wassenburg [4,5], Josefine Axelsson [3], Haiwei Zhang [1], Xianglei Li[6], Jun Hu [7], Hanying Li[1], Liang Zhao[8], Yanjun Cai[1], Youfeng Ning[1] & Hai Cheng [1,6] ✉

Qunf Cave oxygen isotope ($\delta^{18}O_c$) record from southern Oman is one of the most significant of few Holocene Indian summer monsoon cave records. However, the interpretation of the Qunf $\delta^{18}O_c$ remains in dispute. Here we provide a multi-proxy record from Qunf Cave and climate model simulations to reconstruct the Holocene local and regional hydroclimate changes. The results indicate that besides the Indian summer monsoon, the North African summer monsoon also contributes water vapor to southern Oman during the early to middle Holocene. In principle, Qunf $\delta^{18}O_c$ values reflect integrated oxygen-isotope fractionations over a broad moisture transport swath from moisture sources to the cave site, rather than local precipitation amount alone, and thus the Qunf $\delta^{18}O_c$ record characterizes primary changes in the Afro-Asian monsoon regime across the Holocene. In contrast, local climate proxies appear to suggest an overall slightly increased or unchanged wetness over the Holocene at the cave site.

Speleothem oxygen isotope ($\delta^{18}O_c$) records in the Asian summer monsoon domain are important to understand monsoon variability on a wide range of timescales[1–5]. The Asian summer monsoon contains two large monsoon subsystems, the East Asian summer monsoon and the Indian summer monsoon. The speleothem $\delta^{18}O_c$ records from both regions show a broadly similar pattern on the orbital scale, following Northern Hemisphere summer insolation (NHSI), nearly in phase at the precession band (~20-ka cycles, ka: thousand years)[1,6–10], which supports the conventional monsoon hypothesis[11]—the insolation hypothesis[12]. A large number of publications clearly show that the enhanced Asian summer monsoon from the precession minimum ($P_{min}$) or NHSI maximum to the precession maximum ($P_{max}$) or NHSI

minimum, corresponds to the negative $\delta^{18}O_c$ excursions in the speleothem records, and vice versa[1,3,11,13–19]. These data-model comparisons validate the interpretation that speleothem $\delta^{18}O_c$ records from the Asian summer monsoon domain reflect regional Asian summer monsoon intensity or the scale of summer monsoon circulations on orbital scales[1,10]. However, the relation between the speleothem $\delta^{18}O_c$ records and precipitation amount at the cave site may be complex[1–3]. A number of studies suggested that $\delta^{18}O_c$ records from the Asian continent may partially reflect changes in moisture source, isotope fractionation at the source area, transport pathways, as well as the isotopic composition of the Bay of Bengal surface water[13,20–22]. When the Asian summer monsoon is strong, the spatial scale of the summer monsoon

[1]Institute of Global Environmental Change, Xi'an Jiaotong University, Xi'an 710049, China. [2]Department of Environmental Sciences, University of Basel, Basel 4054, Switzerland. [3]Department of Physical Geography and the Bolin Centre for Climate Change, Stockholm University, Stockholm SE-106 91, Sweden. [4]Center for Climate Physics, Institute for Basic Science, Busan 46241, Republic of Korea. [5]Pusan National University, Busan 46241, Republic of Korea. [6]State Key Laboratory of Loess and Quaternary Geology, Institute of Earth Environment, Chinese Academy of Sciences, Xi'an 710061, China. [7]College of Ocean and Earth Sciences, Xiamen University, Xiamen 361102, China. [8]School of Geography, Nanjing Normal University, Nanjing 210023, China. ✉e-mail: cheng021@xjtu.edu.cn

circulation expands, and more remote moisture is transported inland, resulting in lower precipitation oxygen isotope ($\delta^{18}O_P$) values in continental regions and vice versa[1–3,16,23–26]. Theoretically, speleothem $\delta^{18}O_c$, as a proxy of the $\delta^{18}O_P$ at the cave site, is related to oxygen-isotope fractionations integrated from tropical oceans to the cave site along the moisture transport trajectory[25,26]. Besides, the influence of other factors on speleothem $\delta^{18}O_c$, such as the mixing of different moisture sources and rainfall amount, should not be precluded.

The Holocene, from ~11.7 ka BP (before present, where present = 1950 CE) to the present, spans about half of a precession cycle, approximately from $P_{min}$ (NHSI maximum) to $P_{max}$ (NHSI minimum). While a large set of Holocene speleothem records are available in the East Asian summer monsoon domain[3,27,28], similar records in the Indian summer monsoon domain remain sparse and segmented. The well-known Indian summer monsoon record comes from Qunf Cave in Oman, southeastern Arabian Peninsula[29]. The Qunf $\delta^{18}O_c$ record covers most of the Holocene and has been widely used as a typical Indian summer monsoon record. It has been highly cited during the past two decades and remains a unique Holocene $\delta^{18}O_c$ record in the area up till now. Previous studies on the Qunf $\delta^{18}O_c$ record, and additional speleothem records from Oman and Yemen, suggested that these records reflect changes in Indian summer monsoon precipitation amount during the Holocene[29–32]. The sharp decrease of the speleothem $\delta^{18}O_c$ from 10.3 to 9.6 ka BP was interpreted to indicate a rapid northward shift of the summer position of the intertropical convergence zone (ITCZ) and the associated Indian summer monsoon rainfall, leading to a rapid increase in summer monsoon precipitation. After ~8 ka BP, the speleothem $\delta^{18}O_c$ record shows an almost linear response to the orbitally-induced variation in NHSI at 30°N, which implies a gradual southward retreat of the ITCZ and gradual weakening of the Indian summer monsoon in response to a decrease in summer insolation[29]. This interpretation seems to be consistent with the conventional wisdom that increased NHSI enhances the land-sea thermal contrast and, thus, the summer monsoon intensity[11]. However, this classical notion of monsoon has been challenged by a number of recent modeling studies[33–35].

Marine records from the Arabian Sea near Qunf Cave, such as the upwelling-driven biological productivity, foraminifera assemblage, particle size, and oxygen minimum zone records were also interpreted as proxies of the Indian summer monsoon intensity[36–38]. A few marine upwelling records from the Arabian Sea show a consistent Holocene pattern with the speleothem records[39]. However, some marine upwelling records in the Arabian Sea suggest intensified Indian summer monsoon winds throughout the entire Holocene and show a significant lag to NHSI and the Indian summer monsoon intensity change inferred by speleothem $\delta^{18}O_c$ records[36,40–42]. These observations were interpreted to be caused by the latent heat from the southern Indian Ocean and the global ice volume besides insolation[36,42,43]. Alternatively, the apparent lag to the NHSI is interpreted as the length of late summer and the effect of the Atlantic Meridional Overturning Circulation[37,38].

Hsu et al.[44] were the first to show that their climate model actually reproduced this opposite response between land and ocean to NHSI at the precession band. This study was followed by multiple climate model simulations that confirmed the results from refs. 3,16,17,45–47. Later, ref. 2 coined the term "land-sea precession phase paradox" in the research forefront of the Asian summer monsoon, referring to the contrast between speleothem and marine proxy records. Also, Ruddiman[48] argued that the Arabian Sea summer monsoon proxies are not tightly coupled to monsoon intensity but are influenced by other processes. Recently, refs. 45,46 explained the mechanisms for the opposing marine and terrestrial proxy response in the Indian summer monsoon domain. It appears that the upwelling-based proxies in the Arabian Sea are not linked to the Indian monsoon rainfall over India at precession timescales[47]. Changes in the low-level jet could cause a stronger wind stress curl during weaker summer insolation and lead to

strong upwelling. This interpretation appears to largely reconcile the so-called "sea-land precession-phase paradox", as the monsoon wind strength over the oceans is not necessarily coupled with monsoon precipitation over the Asian continent[1,2]. However, in light of these recent discussions and data-model comparisons[16,21,25], critical issues remain regarding: (1) What is the hydroclimate significance of Qunf $\delta^{18}O_c$ variation? (2) What is the local hydroclimate variation pattern across the Holocene? (3) What is the relation between the Qunf $\delta^{18}O_c$ record and NHSI? (4) Is precipitation at Qunf Cave sourced from single or multiple moisture sources? These questions call for further investigations, especially with speleothem multi-proxy data.

In this study, we provide multiple proxies in addition to $\delta^{18}O_c$ from the stalagmite Q5 from Qunf Cave, southern Oman, including $(^{234}U/^{238}U)_0$, $\delta^{13}C$, trace elements, fluid inclusion $\delta D_{fi}$, and $\Delta'^{17}O$ records to reconstruct the local and regional hydroclimate history across the Holocene. We then compare the Qunf Cave record with climate model simulations to evaluate alternative interpretations of Qunf $\delta^{18}O_c$ values.

## Results and discussion
### Climate conditions and chronology

Qunf Cave (17°10′N, 54°18′E; 650 m above sea level), from which stalagmite Q5 was collected, is located in southern Oman (Fig. 1), at the northern fringe of the present summer ITCZ and the northern limit of the Indian summer monsoon. At present, summer monsoon rainfall accounts for more than 90% of the total annual rainfall amount (400–500 mm at the cave site)[29]. The average summer (June, July, and August, JJA) rainfall recorded at the Salalah GNIP station (17°02′N, 54°07′E) is 1.65 mm d$^{-1}$ (1988–1989), and the JJA rainfall of 1989 is 3.43 mm d$^{-1}$ at Qairoon Hairiti GNIP station (17°15′N, 54°05′E). The Somali jet brings considerable moisture to the Qunf Cave site during the summer, mainly from the tropical South Indian Ocean across the equator to the western Arabian Sea (Fig. 1). In comparison with the dry years, the wet years in the area have more local moisture contributions, accompanying with less precipitation over South Asia. Meanwhile, the moisture tagging analysis and simulations also indicate that the Red Sea, the Persian Gulf, Mediterranean and Iranian Plateau provide considerable moisture to the Qunf Cave site (Fig. 1 and Supplementary Fig. 1). Q5 grew from 10.8 to 0.4 ka BP with a hiatus from 2374 (±143) to 840 (±56) year BP (Supplementary Fig. 2), i.e., almost the entire Holocene.

### Moisture sources revealed by multi-proxy records and climate model simulation

Fluid inclusion $\delta D_{fi}$ values provide the isotope composition of the cave drip water from which the speleothem calcite precipitated, and may contain information on the conditions of drip water. Q5 $\delta D_{fi}$ data show a distinct trait. During the Holocene Humid Period in Southern Arabia (~10.5 to ~6 ka BP)[30,32,49,50], $\delta D_{fi}$ values are more depleted, with a mean $\delta D_{fi}$ value of ~2.07‰. After ~6 ka BP, $\delta D_{fi}$ shows an increasing trend, with an average value of ~9.46 ‰ (Fig. 2a). Consistently, the $\delta D_{fi}$ data from Hoti Cave (northern Oman), which sits close to Qunf, also show a significant change in the moisture source, seasonality, and amount of rainfall above the cave around 6 ka BP[49]. Moreover, the slope of the $\ln(\delta^{18}O_c+1)-\ln(\delta^{17}O_c+1)$ line shifts at ~6 ka BP from 0.510 to 0.523 (Supplementary Fig. 4a, b). $\delta^{18}O_c$-$\delta^{17}O_c$ data thus suggest that either moisture source conditions or the moisture sources have changed at ~6 ka BP in southern Oman. Although previous studies based on $\delta^{18}O_c$ records (including Q5) consider that the Indian summer monsoon (or the Indian Ocean) was the sole moisture source in southern Oman[29–31,51], our new data seem to imply an additional moisture source, yet unknown, that could be involved during the Holocene Humid Period[49].

$\Delta'^{17}O$ is defined as the deviation of triple oxygen isotope data from a reference relationship between speleothem $\delta^{17}O_c$ and $\delta^{18}O_c$[52] (See

supplementary text), which is related to distinct hydroclimate dynamics[53–56]. Speleothem Δ'[17]O is not sensitive to Rayleigh distillation from the ocean to the continent[57] (Supplementary Fig. 5a). Therefore, it reflects mainly the original Δ'[17]O signal of water vapor, which in turn allows a calculation of the relative humidity at the moisture source[52,58–60]. In addition, other factors might also contribute to some extent to the Δ'[17]O variation, such as the mixing of water vapors (between the original air masses and continental recycling moisture along air-mass trajectories), re-evaporation of precipitation (depending on the re-evaporation rate and downdraft intensity), and convections[61,62]. Collectively, the process of stronger continental recycling and re-evaporation of precipitation may cause higher Δ'[17]O (Supplementary Fig. 5b). Q5 Δ'[17]O data show higher values (−225 per meg) during -8-6 ka BP compared to early (−259 per meg) and late (−245 per meg) Holocene (Fig. 2b). Between 8 and 6 ka BP, the relative humidity of the moisture source area inferred by Δ'[17]O is lower (-60%) (Fig. 2c). From -6 ka BP to the present, the calculated relative humidity of Q5 moisture source fluctuated around 80%.

To further understand the hydroclimate conditions at Qunf Cave, we compared the EC-Earth simulated summer (JJA) hydrological conditions between 8 K (8 ka BP) and PI (pre-industrial period). Our model simulations indicate that during the 8 K period, the North African summer monsoon−characterized by the southwesterly wind extending across the continent and reaching the southern Arabian Peninsula was considerably stronger (Fig. 3a). Although it is possible that the -900-m high Dhofar mountain could prevent some of the moisture from the African continent to reach the Qunf Cave area, both the summer monsoon wind from the Arabian Sea in the south and the

African continent in the west could have contributed water vapor to southern Oman at 8 K. This is in line with an expansion of the North African summer monsoon across the Red Sea during the high NHSI time[16,63]. The model simulations reveal a -6% increase in relative humidity at the equatorial Atlantic, Arabian Sea and Qunf Cave site from 8 K to PI (Fig. 3b). The Δ'[17]O-based reconstruction of relative humidity at the moisture source shows a -15% increase from middle to late Holocene (Fig. 2c). The direction of these Holocene trends is consistent. Taking into account the uncertainties of the modeled and reconstructed relative humidity, these datasets show strong similarities.

The influence of the Indian summer monsoon and its interaction with the North African summer monsoon may not be ignored as the Arabian Sea is located right in the confluence region between African and Asian summer monsoon systems[13,16,63,64]. The North African summer monsoon has effective control over the Eastern Mediterranean deepwater ventilation on orbital time-scales through changing the discharge of the Nile River. This results in density stratification and breakdown of deepwater formation reflected by sapropel layers S1 in Ocean Drilling Program Site 968 from -10.2 to 6.5 ka BP[65–68]. This strong North African summer monsoon period corresponds precisely with the lowest δ[18]O_c period in the Q5 record (Fig. 4). Furthermore, the Soreq and Jeita cave δ[18]O_c records from the Levant are also sensitive to changes in δ[18]O of the Eastern Mediterranean surface seawater. More negative δ[18]O_c values are indicative of higher Nile discharge from the early to middle Holocene[69,70] (Fig. 4). Additionally, δD_wax records from eastern Africa, which reflect North African summer monsoon variations, also exhibit a trend similar to the Q5 δ[18]O_c record[71] (Fig. 4). After

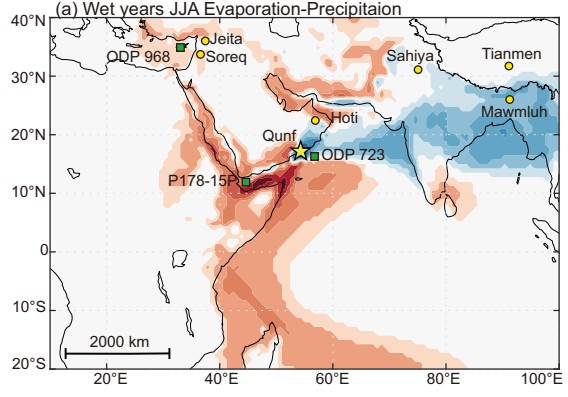
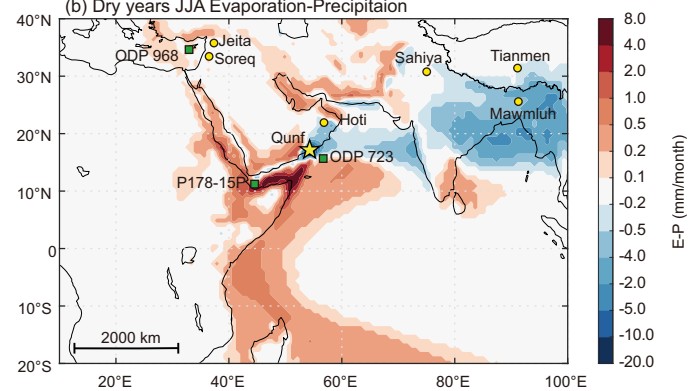
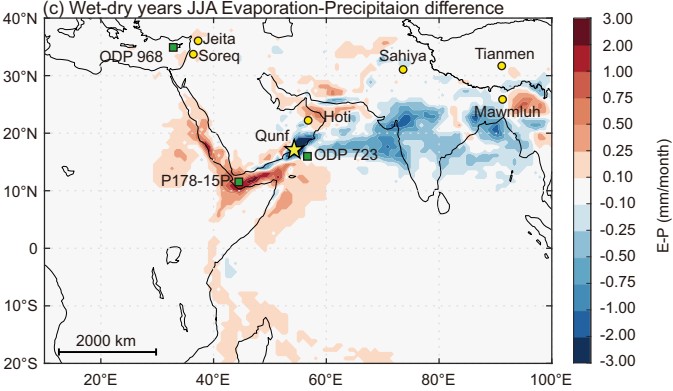

**Fig. 1 | Qunf Cave (yellow asterisk) June–July–August (JJA) evaporation minus precipitation (E-P) diagnosed from 12-days back trajectories using the FLEX-PART. a** Results for the highest precipitation (>+1.5σ) years (1983, 1986 and 1996). **b** Results for the lowest precipitation (<−1.5σ) years (1999, 2000, and 2012). **c** The difference between the wet and dry years. Positive values (red) indicate a larger net moisture supply, while negative values (blue) indicate higher water vapor condensation/precipitation. Sites of other caves (yellow circles) and marine sediment cores (green boxes) that contain important Holocene paleoclimate records (discussed in the text) are also shown.

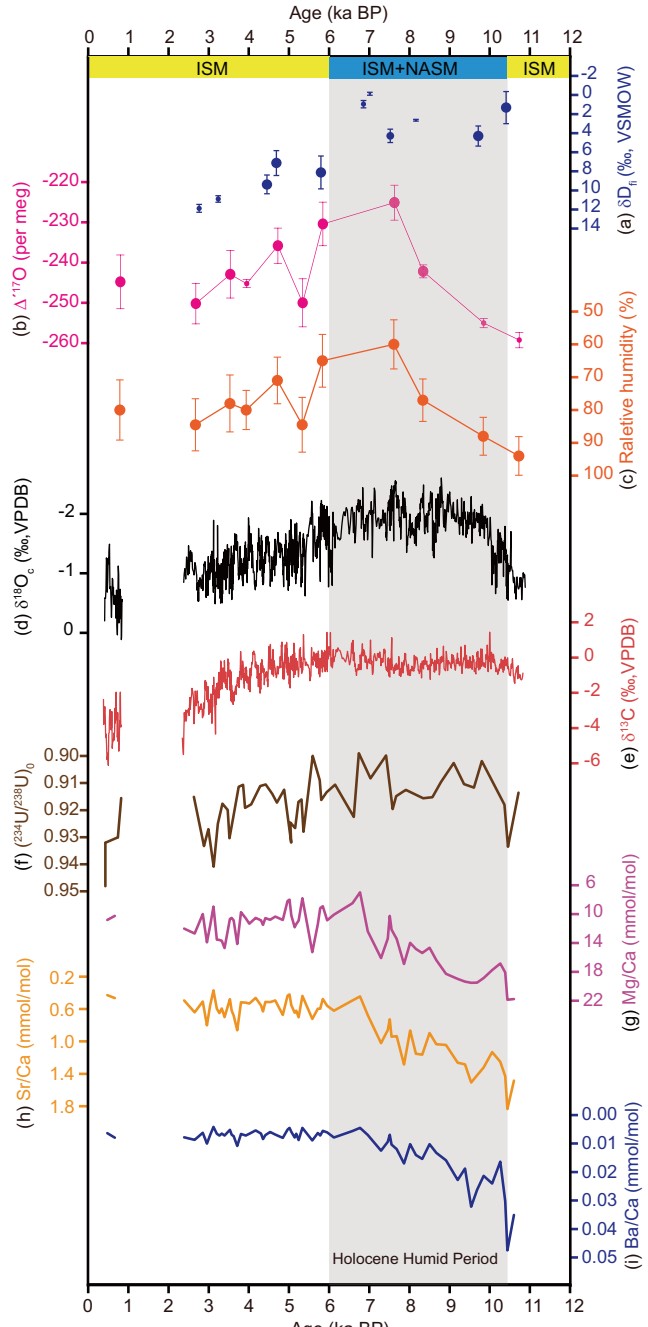

**Fig. 2 | Q5 multi-proxy records. a** $\delta D_{fi}$, error bars are the standard deviation (1 SD) of the reproducibility of crushed speleothem samples; **b** $\Delta'^{17}O$, error bars are the standard deviation (1 SD); **c** Relative humidity calculated by Q5 $\Delta'^{17}O$ data, error bars are the standard deviation (1 SD); **d** $\delta^{18}O_c$; **e** $\delta^{13}C$; **f** $(^{234}U/^{238}U)_0$; **g** Mg/Ca; **h** Sr/Ca; and **i** Ba/Ca. The top yellow and blue banners show the monsoon systems prevailed at the cave site, with ISM stands for Indian summer monsoon, and NASM stands for North African summer monsoon. The Holocene Humid Period is from 10.5 to 6 ka BP, coinciding with the intensified NASM and ISM[49]. Source data are provided in the Source Data file.

~6 ka BP, the North African summer monsoon fringe retreated from the southern Arabian Peninsula/northeastern Africa associated with the termination of the Mediterranean sapropel S1 period and the termination of the Holocene African Humid Period (Fig. 4). The Qunf multi-proxy datasets, combined with a set of model simulations on the Holocene hydroclimate evolution in the region, allow for a thorough comparison between observed and modeled Holocene hydroclimate

conditions. We propose that there was a large-scale reorganization of atmospheric circulation in the region. In addition to the water vapor transported by the Indian summer monsoon from the adjacent Arabian Sea, the North African summer monsoon may have contributed remote moisture from the tropical Atlantic via Northeast Africa to Qunf Cave during the early and middle Holocene. This finding offers robust evidence supporting earlier model results[16,17]. In this regard, the shift in $\delta D_{fi}$ and $\Delta'^{17}O$ data around 6 ka BP could indicate a decrease/withdrawal in the contribution of the North African summer monsoon[63].

## Understanding of Qunf $\delta^{18}O_c$ record
$\delta^{18}O_c$ values of stalagmite Q5 show an abrupt decrease (from −0.2 to −2.5‰) that occurred from ~10.8 to 9 ka BP, followed by a relatively stable period (~ −2‰) between 9 and 6 ka BP, and then by a persistent increase of $\delta^{18}O_c$ from −2‰ to ~ 0‰ till a hiatus from ~2.4 to 0.8 ka BP. After the hiatus, $\delta^{18}O_c$ decreased from ~ 0‰ to ~−1‰ over ~400 years (Fig. 2d). $\delta^{18}O_c$ shows the same trends as $\delta D_{fi}$, supporting that the $\delta^{18}O_c$ reflects drip water $\delta^{18}O$ variability. Currently, the effect of the Indian summer monsoon is prevailing over the southern Arabian Peninsula (Fig. 1). The long-term increasing trend of the Qunf $\delta^{18}O_c$ record from ~6 ka BP to the present may indicate an overall Indian summer monsoon (or summer monsoon circulation) waning process[8,49,72]. A significant mid-Holocene positive shift of $\delta^{18}O_c$ at ~6 ka BP is also found in Hoti Cave in northern Oman (Fig. 4), marking the southward retreat of the ITCZ and the associated Indian summer monsoon rainfall belt and a change in the seasonality of rainfall[32,49]. As aforementioned, water vapor from the equatorial Atlantic could be transported across the African continent by strong North African summer monsoon wind to Qunf Cave during the early to middle Holocene. Thus, from ~10 to ~6 ka BP, southern Oman was likely affected by both the Indian summer monsoon and likely, but to a lesser extent, the North African summer monsoon, both resulting in the more negative $\delta^{18}O_P$ owing to larger-scale summer monsoon circulations and associated remote moisture[1–3], as well as the increased local precipitation amount effect ("amount effect")[29–31].

Taking a broader perspective, the monsoons over the African continent and the Indian Ocean exhibit remarkably similar hydroclimate trends during the middle to late Holocene. This suggests that they were sensitive to, or driven by, a common forcing mechanism on suborbital timescales[73,74]. More broadly, the entire North African summer monsoon and Asian summer monsoon (Indian summer monsoon and East Asian summer monsoon) may be viewed as a large unified summer monsoon system. In this interpretative framework, the Qunf $\delta^{18}O_c$ record shows the Holocene hydroclimate variation of this "super-monsoon" or Afro-Asian monsoon regime[63,64], since the Qunf $\delta^{18}O_c$ variations appear to follow the overall changes of the Afro-Asian monsoon intensity according to both empirical and model results as aforementioned.

The Qunf $\delta^{18}O_c$ record mostly reflects large-scale monsoon circulation changes, making local hydroclimate proxies an essential complement. Combining the $(^{234}U/^{238}U)_0$, $\delta^{13}C$, and trace element records from Q5 (Fig. 2) with the simulation results (Fig. 3 and Supplementary Fig. 7) enables us to reconstruct local precipitation minus evaporation (P-E) over the Holocene. Our simulation results show that both the North African summer monsoon and Indian summer monsoon were much stronger at 8 K than at PI. Notably, the summer precipitation amount at Qunf Cave was higher at 8 K. This may lead to increased water flow rates in Qunf Cave's epikarst, resulting in lower $(^{234}U/^{238}U)_0$ values, as $(^{234}U/^{238}U)_0$ is influenced by water flow rates and water−rock interaction times[75–77].

However, the summer P-E appears to be slightly lower over a large part of the Arabian Peninsula, suggesting a strong evaporation condition (Fig. 3, Supplementary Fig. 7). These results are consistent with Q5 $\delta^{13}C$ and trace elements ratio data. Lower summer temperature and

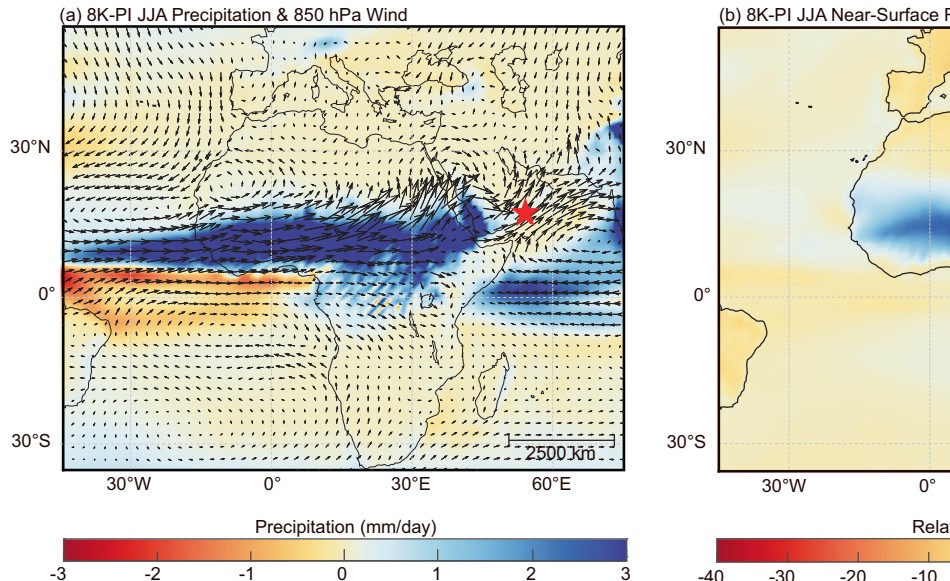

**Fig. 3 | EC-Earth simulation results of June–July–August (JJA) hydroclimate difference between 8 ka BP (8 K) and preindustrial period (PI). a** Precipitation (shadings) and 850hPa wind (vectors) difference. **b** Near-surface relative humidity difference. The red asterisk shows the location of Qunf Cave. All the results have passed the significance test of the mean difference.

slightly increased P-E could favor increased organic materials and reduced prior calcite precipitation (PCP), resulting in more negative $\delta^{13}C$ values in the late Holocene[78–80]. Q5 $\delta^{13}C$ values range from -6.4‰ to 1.1‰, with relatively stable values (~ −1‰) from -10.8 to 6 ka BP and a decreasing trend from−~1‰ to ~−5‰ since ~6 ka BP (Fig. 2), suggesting slightly lower P-E (or effective wetness) in the early and middle Holocene compared to the late Holocene. Trace element ratios (Mg/Ca, Sr/Ca, and Ba/Ca) show a significant decreasing trend from 10.8 to ~6 ka BP, followed by low and stable ratios after ~6 ka BP (Fig. 2). PCP often occurs in the vadose zone of many cave systems, and increases with reduced effective infiltration, affecting trace element ratios in cave drip waters[81–83]. The slightly lower P-E condition at 8 K (or the early to middle Holocene, Supplementary Fig. 7) suggests increased evaporation under higher summer insolation, promoting PCP and resulting in observed higher trace element ratios. The low and stable trace element ratios after ~6 ka BP could be due to the expected trend of reduced summer evaporation caused by decreased NHSI from the early/middle to late Holocene.

Both model simulation and proxy records show a similar increasing trend in local hydroclimate conditions at the cave site from 8 K to PI, the P-E was lower during the early to middle Holocene compared to the late Holocene (See supplementary text). However, given the variety of factors and multiplicity of proxy interpretations, further studies remain a prerequisite to understanding the Holocene hydroclimate history in the region.

### The Indian summer monsoon variations across the Holocene

The understanding of monsoons as an integral component of the global atmospheric circulation and hydroclimate is becoming prevalent[33,35]. Of note are three primary theoretical concepts of monsoons: one based on convective quasi-equilibrium (CQE)[84,85], another founded on the moist static energy budget[86,87], and a third that frames the monsoon as an extension of the zonal-mean ITCZ[34,35]. While these theoretical frameworks help explain certain aspects of modern monsoon variability, especially regarding monsoon rainfalls and thermodynamics, the interpretations of paleoclimate variations across various timescales are still in development[45,46,88,89].

Physically, the summer barometric differentials and other boundary conditions on a large spatial-scale, regardless of their cause, drive the Asian summer monsoon circulation (e.g., at 850 hPa) (Supplementary Fig. 8) and the associated large-scale vapor flux (Supplementary Fig. 9). In terms of precipitation $\delta^{18}O_P$ at the cave site, prior to the onset of the Asian summer monsoon, the moisture source is relatively proximate, resulting in heavier $\delta^{18}O_P$. After the summer monsoon onset, the spatial-scale of monsoon circulation and moisture from remote sources increase dramatically (Supplementary Figs. 8 and 9), both leading to lighter precipitation $\delta^{18}O_P$ at the cave site. Consequently, cave $\delta^{18}O_C$ mainly indicates the dynamic aspect of the monsoon circulation[90], with a small portion of the variability explained by local rainfall amount or thermodynamics. Local rainfall amount or P-E is, however, better reflected by other proxies [e.g., trace elements, $(^{234}U/^{238}U)_0$, and $\delta^{13}C$].

As the Indian summer monsoon is an interhemispheric monsoon system, the interhemispheric differential in tropical insolation (or Summer Inter-Tropical Insolation Gradient, SITIG[91]) is critical, reflecting both "pull" and "push" forcings from Northern and Southern Hemisphere that propel monsoon changes[1]. Thus, we use the interhemispheric differential in tropical insolation (30°N-30°S) as an integrated insolation forcing of the monsoon, consistent with previous evidence, such as the Asian summer monsoon variation pattern over the MIS 3[1]. The Qunf $\delta^{18}O_C$ record suggests an Indian summer monsoon waning trend from the middle Holocene to the present, broadly following SITIG. This observation is consistent with many other Indian summer monsoon records, such as the Sahiya and Mawmluh records from northern India[72,92], the Tianmen record from the southern Tibetan Plateau[93], and some of the upwelling records from the Arabian Sea and its adjacent area[39] (Fig. 4).

In the early 1990s, ref. 36 studied a set of upwelling proxies from marine sediment cores from the Arabian Sea. They raised a hypothesis that changes in the Indian summer monsoon were driven not only by NHSI, but also by the latent heat release of moisture originating from the southern Indian Ocean and the global ice volume affecting the Indian summer monsoon precession phases. Sea surface temperature variation over the south subtropical Indian Ocean controls the latent heat effect on the Indian summer monsoon, which has a significant phase lag to the NHSI maxima at the precession band[36,40,94–96]. Multiple upwelling records indeed show a significant phase lag (up to 8–11 ka) or nearly anti-phase to NHSI at the precession band[40–42]. Model

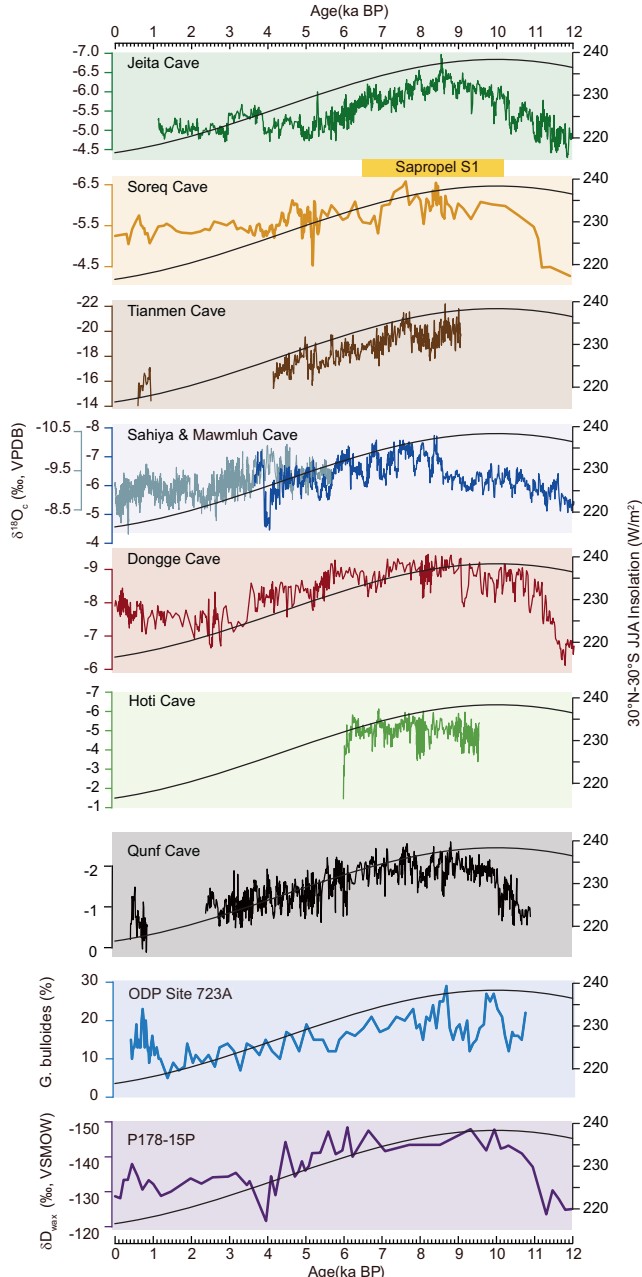

**Fig. 4 | Comparison of the Q5 $\delta^{18}O_c$ record with other North African summer monsoon and Asian summer monsoon proxy records, and 30°N–30°S June–July–August (JJA) insolation.** The sapropel S1 from the Eastern Mediterranean Ocean Drilling Program (ODP) Site 968 is shown between 10.2 and 6.5 ka BP (marked as a yellow rectangle)[65]. From top to bottom: $\delta^{18}O_c$ of Jeita Cave[70] and Soreq Cave[69] from the Levent region, $\delta^{18}O_c$ of Tianmen Cave[93] from Tibetan Plateau, $\delta^{18}O_c$ of Mawmluh Cave[81] (blue) and Sahiya Cave[72] (gray) from India, $\delta^{18}O_c$ of Dongge Cave from southern China[117], $\delta^{18}O_c$ of Hoti Cave from northern Oman[31], $\delta^{18}O_c$ of Qunf Cave (this study), *G. bulloides* percentage in ODP Site 723 A offshore Oman[39], and $\delta D_{wax}$ of Horn of Africa marine record P178-15P[71]. Source data are provided in the Source Data file.

simulations reveal contrasting hydroclimatic responses between land and ocean to NHSI changes at the precession band[16,17,44]. This phenomenon has been referred to as the "land-sea precession phase paradox"[2]. In the Indian summer monsoon regions, Jalihal et al.[46,47,97] investigated the opposing marine and terrestrial responses, both empirically and theoretically. They discovered that, in addition to NHSI, precipitation changes over the ocean are influenced by changes

in surface energy fluxes and vertical stability. This view largely reconciles the apparent discrepancy between the Qunf $\delta^{18}O_c$ record and some previous marine upwelling records from the Arabian Sea in terms of the precession phase paradox.

Overall, our speleothem multi-proxy records and modeling results provide valuable insights into the interpretation of the Qunf $\delta^{18}O_c$ record and the Holocene hydroclimate history in the southern Arabian Peninsula. During the early to middle Holocene, corresponding to the Holocene Humid Period in southern Arabia, the North African summer monsoon, in addition to the Indian summer monsoon, may likely contribute water vapor to the Qunf Cave area as well. After ~6 ka BP, Qunf $\Delta^{17}O$ data suggest a possible reduction in moisture delivered by the North African summer monsoon, higher $\delta^{18}O_c$ values suggest that only the Indian summer monsoon prevails with water vapor mainly derived from the proximate Arabian Sea like today. Our results are consistent with the interpretative framework that in addition to the amount effect, speleothem $\delta^{18}O_c$ values are essentially related to oxygen-isotope fractionations integrated during the moisture transport from the source to the cave site. More broadly, the Qunf $\delta^{18}O_c$ record may be viewed theoretically as an archive indicating the change of the vast monsoon regime, the unified North African summer monsoon - Asian summer monsoon system. Although the Qunf $\delta^{18}O_c$ have lighter values and the precipitation amount might increase from ~10 to ~6 ka BP, Qunf $(^{234}U/^{238}U)_0$, $\delta^{13}C$, and trace element ratio records (Mg/Ca, Sr/Ca, Ba/Ca) seem to suggest a lower P-E condition. This requires further investigation.

## Methods

### Modern source–receptor relationship

To establish a modern source–receptor relationship for moisture transport in the study region, the Lagrangian trajectory model FLEX-PART, driven by ERA-Interim reanalysis, was used. Based on ERA-interim precipitation (1979–2019), we created wet and dry composites from the three driest ($-1.5\sigma$) and wettest ($+1.5\sigma$) June-July-August (JJA) years in our study area (17°N–17.75°N, 54°E-54.75°E). FLEXPART simulations were performed during JJA in these years.

The FLEXPART simulation was driven by ERA-interim through 6-hourly analyzes (at 00.00, 06.00, 12.00, and 18.00 UTC) and 3-hourly forecasts at intermediate times (at 0300, 0900, 1500, and 2100 UTC), with 1° × 1° resolution on 60 model levels[98]. Surface moisture flux is calculated over an area ($A$), where E-P for the total particles residing over $A$ is given by:

$$E - P \approx \frac{\sum_{k=1}^{K}(e - p)}{A} \qquad (1)$$

where $e$-$p$ is the rate of moisture change along the trajectory[99]. The simulation releases 50,000 particles, and $K$ is the number of $N$ particles that resides over $A$. This approach divides the whole atmosphere's mass into small elements distributed homogeneously in the atmosphere according to the atmospheric mass distribution. These are then moved with the mass-consistent winds and are based on mass-consistent turbulence and convection parameterizations[100,101]. We calculated the mean net moisture flux based on E-P and backtracked elements that had a loss (E-P < 0) over the Qunf Cave region (17°N –17.75°N, 54°E-54.75°E) for 12 days.

### $^{230}$Th dating

A total of 50 new $^{230}$Th dates were obtained from stalagmite Q5. Powder sub-samples were hand-drilled parallel to the growth band near the stalagmite axis. The $^{230}$Th dating work was performed at Xi'an Jiaotong University Isotope Laboratory using multi-collector inductively coupled plasma mass spectrometers (Thermo-Finnigan Neptune-*plus*). We used standard chemistry procedures to separate U and Th for dating[102]. A triple-spike ($^{229}$Th–$^{233}$U–$^{236}$U) isotope dilution

method was employed to correct instrumental fractionation and determine U-Th isotopic ratios and concentrations. Details about the instrumental setup are provided by refs. [103,104]. All ages are in stratigraphic order within dating uncertainties. The age model was constructed using the Constructing Proxy Records from Age (COPRA) program[105]. Q5 $\delta^{18}O_c$ and $\delta^{13}C$ records were adjusted to the new age model (Supplementary Fig. 2).

## Trace element analysis

The trace elements were measured by inductively coupled plasma atomic emission spectroscopy (ICP-AES) (Thermo Scientific iCAP DUO 6300) in Instituto Pirenaico de Ecologia, Spanish Scientific Research Council (IPE-CSIC). The drilled powder (100–150 mg) was placed in tubes cleaned with 10% HCl, rinsed with MilliQ-filtered water, and dissolved in 1.5 mL of 2% $HNO_3$ (Tracepur) immediately before analysis. Samples were run at average Ca concentrations of 200 ppm. Calibration was conducted offline using the intensity ratio method described by ref. [106]. The results are shown by molar ratios of Mg, Ba, and Sr relative to Ca.

## Triple oxygen isotope analysis

The $\delta^{17}O_c$ of Q5 was measured at Xi'an Jiaotong University Isotope Laboratory. After drilling the carbonate powder of the sample, we add in phosphoric acid ($H_3PO_4$, 1.92 g/ml, -104‰) at 25 °C to extract $CO_2$. Then the $CO_2$ was equilibrated with an equal amount of the $O_2$ gas for 30 min under 750 °C to reach a Pt-catalyzed equilibrium by an $O_2$–$CO_2$ Pt-catalyzed oxygen-isotope equilibration reaction system. In this system, the two post-equilibration gases were separated from each other cryogenically. Finally, $\delta^{17}O_c$ was obtained through measurements of the resultant $O_2$ and $CO_2$ by a Thermo Scientific MAT 253 mass spectrometer[60]. The calculation of $\Delta^{17}O$ and the relative humidity at the moisture source follows the method from ref. [60] (See supplementary text).

## Fluid inclusion measurements

Q5 $\delta D_{fi}$ was analyzed at Xi'an Jiaotong University Isotope Laboratory. The Q5 calcite blocks were crushed at a temperature of -120 °C. The liberated water was then transported to a wavelength-scanned cavity ring-down spectroscopy system (Picarro L2140-i analyzer). The analytical method is described in ref. [107], the maximum error on the reproducibility of crushed speleothem samples for this system is 2‰ for $\delta D_{fi}$ (1 SD). $\delta D_{fi}$ values are reported on the Vienna Standard Mean Ocean Water to SLAP2 (VSMOW2 – SLAP2) scale.

## Model simulations

The simulations based on the fully coupled Earth system model EC-Earth are used in this study. A consortium of European research institutions develops EC-Earth to build a fully coupled Atmosphere-Ocean-Land-Biosphere Earth system model for seasonal to decadal climate prediction and future climate projections[108]. The atmospheric model of EC-Earth is based on the IFS, including a land model H-TES-SEL, developed at the European Centre for Medium-Range Weather Forecasts (ECMWF)[98]. In addition, the dynamic vegetation model LPJ-GUESS[109] is coupled to the land component. The ocean component is the Nucleus for European Modelling of the Ocean (NEMO)[110] and includes a sea-ice model LIM3[111]. We use the CMIP6 configuration of EC-Earth3-veg-LR, in which the atmosphere and land model has a T159 horizontal spectral resolution (roughly 1.125°–125 km) with 62 vertical levels. The NEMO and LIM have a nominal horizontal resolution of 1°, and NEMO has 75 vertical levels. The coupling between the atmosphere and ocean/sea ice is through the Ocean Atmosphere Sea Ice Soil coupler (OASIS, version 3.0)[112].

We simulate the 8 ka BP (8 K) and a pre-industrial control simulation (PI) using EC-Earth3-veg-LR. The simulations follow the CMIP6-PMIP4 protocol for each experiment setup[113]. The PI and 8 K simulations have the same boundary conditions except for the orbital forcing and greenhouse gas concentration differences (Supplementary Table 1). The orbital forcing is computed in the IFS component according to ref. [114], as described in ref. [115]. The greenhouse gas concentration for PI follows CMIP6 and for 8 K using the reconstruction from ref. [116].

The initial condition for PI simulation with EC-Earth-veg-LR is taken from a previous PI run with EC-Earth3-LR[115]. The simulation reaches the quasi-equilibrium after 300 years (meets the criteria global mean surface temperature trend <±0.05 K per century[113], and the simulation continues running for another 700 years. The initial condition for the 8 K simulation is taken from the equilibrium state of PI simulation, with changed orbital forcing and Green House Gas concentration. For the 8 K condition it takes around 300 years to reach equilibrium, and we run for another 700 years. The last 200 years' outputs are used to analyze both simulations.

## Data availability

Source data for Fig. 2 and Fig. 4 are referenced in the Source data provided with this paper. The absolute $^{230}Th$ dates and multi-proxy data time series are available online on the NOAA paleoclimate database (https://www.ncei.noaa.gov/access/paleo-search/study/38279). FLEXPART model and documentation can be found at https://www.flexpart.eu/ (last accessed: 2023-03-09). Climate model simulation data is published on Zenodo (https://zenodo.org/record/8121547). Source data are provided with this paper.

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

## Acknowledgements

This work was supported by the National Natural Science Foundation of China (NSFC grants 41888101, 42150710534 to H.C.; grant 4197020225 to H.Z.), Q.Z. acknowledges support from the Vetenskapsrådet (grant nos. 2013-06476, 2017-04232), J.A.W. acknowledges support from the Institute for Basic Science (IBS), Republic of Korea, under IBS-R028-Y2. The EC-Earth simulations were performed using ECMWF's computing and archive facilities and the Swedish National Infrastructure for Computing (SNIC) at the National Supercomputer Centre (NSC). Y.T. thanks Zhongli Liu for his support when he was at Xi'an Jiaotong University and now at Peking University.

## Author contributions

Y.T. wrote the draft manuscript. Y.T. and H.C. designed the research and experiments. Y.T. performed the fluid inclusion measurements. D.f. collected the speleothem sample and provide the stable oxygen and carbon isotope records. Q.Z. conducted EC-Earth simulations. L.S. performed triple oxygen measurements. J.A.W. contributed to the fluid inclusion and the local hydroclimate analysis. J.A. ran the FLEXPART analysis. Y.T., X.L., and Y.N. contributed to $^{230}Th$ dating. Y.T., X.L., and H.L. did the trace element measurements. J.H. and L.Z. contributed to the simulation analysis. Y.T., D.F., Q.Z., L.S., J.A.W., J.A., H.Z., X.L., J.H., H.L., Y.C., L.Z., Y.N., and H.C. reviewed and provided revisions for the manuscript.

## Competing interests

The authors declare no competing interests.
