## [Peer Review File · Nature Communications]

Holocene climate change in southern Oman deciphered by speleothem records and climate model simulationsREVIEWER COMMENTS

Reviewer #1 (Remarks to the Author):

Reviewer's comments on "New insights into speleothem oxygen isotope deciphered by multi-proxy and climate model simulations in southern Oman" Ye Tian et al.

General comments

Tian et al. in this manuscript, have attempted to interpret multiple proxies taken from a cave in Oman, in the context of Indian monsoon over the Holocene. They have tried to substantiate and elucidate their results with climate model simulations. They find that the source of moisture changed across the Holocene, due to changes in large scale monsoon circulation. While interesting, I do not find this result to be novel enough to justify publication in a high impact journal and warrants a rejection. I find a bit of ambiguity/lack of clarity in some of the sentences which allows for a benefit of doubt. Hence, I want to give the authors a chance to respond to the following major concerns, and I recommend a major revision instead.

Major concerns

1. Novelty of the work: Battisti et al. 2014, have already established that Qunf cave proxy does not represent precipitation intensity. It merely captures the isotope fractionation of water vapor - implying the source of moisture as the driver of $\delta^{18}O$. A number of recent literature (eg. Dayem et al. 2010, Krklec et al. 2014, Kathayat et al. 2020; to name a few) have demonstrated that isotope fractionation of oxygen from speleothems are related to changes in the source of moisture. I find the current study one of the many of this growing body of literature on this aspect.

2. Reconstruction of Humidity: The authors have calculated humidity based on $\Delta^{17}O$ and temperature from a transient simulation TraCE-21k. One should be careful while combining parameters from proxy and a climate model. Moreover, it is not clear, how the temperature was used in their equation. The authors should also show RH calculated entirely from the output of TraCE-21k. Does RH calculated from TraCE-21k match the reconstructed RH?

3. Interpretation based on relative humidity: The authors have shown that RH in the proxy was different in early Holocene compared to the late Holocene. This can be due to either a change in local RH or a change in source of moisture or a combination of these two factors. Since, there is no definite way to isolate these two effects on the proxy, the authors have used EC-Earth simulations. It is a well known fact that most CMIP models simulate a constant RH (Eg. Gormon & Muller 2010, Allan et al. 2020, Douville et al. 2022). Consistent with this, EC-Earth shows no changes in RH, except over North African summer monsoon region and the Arabian peninsula.

This is primarily due to the fact that these regions are deserts in the control. The large change in the hydrology during the early Holocene leads to large change in RH, in these regions. The model shows an increase in RH as expected. However, figure 2c suggests that the proxy has a recorded a decrease in RH. The direction of change suggested by the proxy and model are not consistent. This inconsistency has not been discussed. A change in moisture source surely cannot have such a drastic impact. Even if I assume the authors interpretation is correct, Qunf is too close to the Arabian Sea (roughly 15 km) - I'd still say that moisture from the Arabian Sea dominates. In fact, a visual examination of figure 3b suggests that almost entire moisture and winds into the cave site are from the Arabian Sea.

Suggestion: use FLEXPART to identify the source. Or the tagging technique used in Kathayat et al. 2020.

4. Use of FLEXPART The authors have used FLEXPART in the modern context only. They have identified moisture sources for precipitation at the site of the cave based on modern data.

They have used two yrs - one with lowest and another highest annual precipitation in modern climate. For more robustness I recommend that the authors instead use a composite of high and low rainfall years (defined by precip greater than or lower than 1 sigma for example). As acknowledged by the authors, the sources of moisture change across the Holocene. Still, the authors use results based on modern day FLEXPART analysis to interpret their proxy. I suggest that the authors do the analysis with the 8ka and PI simulations to make a strong case.

5. Insufficient literature Line 85-88 and Lines 221 - 225: I have been following the recent developments in the debate between marine and terrestrial proxies of monsoons quite closely. This debate has been largely resolved recently. Therefore, it is important for the benefit of the readers to have a couple of additional sentences. This will help the readers get a perspective of why it is important to isolate processes in the Arabian Sea and those related to Qunf - which is quite essential for this paper. The following papers must be included: Hsu et al 2010 was the first to suggest the opposing response of land and ocean monsoons, followed by Battisti et al. 2014, Bosmans et al. 2018. Jalihal et al. 2019 & Jalihal et al. 2020, were the first papers to connect it to the opposing marine and terrestrial proxy response, and provide a mechanism for the opposite response. Later, Cheng et al, 2021 coined the term "land-sea precession phase paradox".

Minor points

1. Obsolete theory of monsoon: Line 69-70: The authors refer to the enhancing land-sea thermal contrast as major cause of stronger monsoon. Over the last couple of decades, studies have discredited the land-sea thermal contrast theory of monsoons and moist static energy based theories have gained momentum (see review papers eg. Gadgil 2018, Biasutti et al. 2018, Hill 2020). This has been used to understand orbital-scale monsoon variability (D'Agostino et al. 2019, 2020, Jalihal et al 2019, 2020). Accordingly text needs to be modified at appropriate places eg. lines 210-213.

2. Line 35: Include citations to Mohtadi et al. 2014, Seth et al. 2019

3. Line 53: As noted later in the literature, it is not just the fractionations integrated from tropical oceans to cave site, but mixing from other moisture sources etc. It will be good to mention these factors in the introduction itself.

4. Line 101: ISMt -> ISM

5. Line 166-168: How is this statement different from the main conclusion of Battisti et al. 2014? Perhaps, a more correct statement would be that the authors have an observational evidence for a result proposed by Battisti et al. based on models.

6. Line 174-175: "Currently, the ISM is prevailing along the southern Arabian Peninsula". This statement is wrong. How can Indian summer monsoon be prevalent over the Arabian peninsula? I think the authors mean, the effect of ISM is prevalent over the Arabian peninsula.

References:

Allan, R.P., Barlow, M., Byrne, M.P., Cherchi, A., Douville, H., Fowler, H.J., Gan, T.Y., Pendergrass, A.G., Rosenfeld, D., Swann, A.L. and Wilcox, L.J., 2020. Advances in understanding large-scale responses of the water cycle to climate change. *Annals of the New York Academy of Sciences*, 1472(1), pp.49-75.

Battisti, D., Ding, Q., and Roe, G., 2014. Coherent pan-Asian climatic and isotopic response to orbital forcing of tropical insolation, *J. Geophys. Res.-Atmos.*, 119, 11997-12020.

Biasutti, M., Voigt, A., Boos, W.R., Braconnot, P., Hargreaves, J.C., Harrison, S.P., Kang, S.M., Mapes,

B.E., Scheff, J., Schumacher, C. and Sobel, A.H., 2018. Global energetics and local physics as drivers of past, present and future monsoons. *Nature Geoscience*, 11(6), pp.392-400.

Bosmans, J.H.C., Erb, M.P., Dolan, A.M., Drijfhout, S.S., Tuenter, E., Hilgen, F.J., Edge, D., Pope, J.O. and Lourens, L.J., 2018. Response of the Asian summer monsoons to idealized precession and obliquity forcing in a set of GCMs. *Quaternary Science Reviews*, 188, pp.121-135.

D'Agostino, R., Bader, J., Bordoni, S., et al., 2019. Northern Hemisphere monsoon response to mid-Holocene orbital forcing and greenhouse gas- induced global warming. *Geophys. Res. Lett.* 46(3), 1591-1601.

D'Agostino, R., Brown, J. R., Moise, A., et al., 2020. Contrasting Southern Hemisphere Monsoon Response: MidHolocene Orbital Forcing versus Future Greenhouse Gas-Induced Global Warming. *Journal of Climate*, 33(22), 9595-9613.

Dayem, K.E., Molnar, P., Battisti, D.S. and Roe, G.H., 2010. Lessons learned from oxygen isotopes in modern precipitation applied to interpretation of speleothem records of paleoclimate from eastern Asia. *Earth and Planetary Science Letters*, 295(1-2), pp.219-230.

Douville, H., Qasmi, S., Ribes, A. and Bock, O., 2022. Global warming at near-constant tropospheric relative humidity is supported by observations. *Communications Earth Environment*, 3(1), pp.1-7.
Gadgil, S., 2018. The monsoon system: Land–sea breeze or the ITCZ?. *Journal of Earth System Science*, 127(1), pp.1-29.

Hill, S.A., 2019. Theories for past and future monsoon rainfall changes. *Current Climate Change Reports*, 5(3), pp.160-171.

Hsu, Y.-H., Chou, C., and Wei, K.-Y., 2010. Land–ocean asymmetry of tropical precipitation changes in the mid-Holocene. *J. Climate*, 23, 4133–4151.

Jalihal, C., Bosmans, J.H.C., Srinivasan, J. and Chakraborty, A., 2019. The response of tropical precipitation to Earth's precession: the role of energy fluxes and vertical stability. *Climate of the Past*, 15(2), pp.449-462.

Jalihal, C., Srinivasan, J. and Chakraborty, A., 2019. Modulation of Indian monsoon by water vapor and cloud feedback over the past 22,000 years. *Nature communications*, 10(1), pp.1-8.

Jalihal, C., Srinivasan, J., & Chakraborty, A., 2020. Different precipitation response over land and ocean to orbital and greenhouse gas forcing. *Scientific Reports*, 10(11891), 1– 11

Kathayat, G., Sinha, A., Tanoue, M., Yoshimura, K., Li, H., Zhang, H. and Cheng, H., 2021. Interannual oxygen isotope variability in Indian summer monsoon precipitation reflects changes in moisture sources. *Communications Earth Environment*, 2(1), pp.1-10.

Krklec, K. and Domínguez-Villar, D., 2014. Quantification of the impact of moisture source regions on the oxygen isotope composition of precipitation over Eagle Cave, central Spain. *Geochimica et cosmochimica acta*, 134, pp.39-54.

Mohtadi, M., Prange, M., and Steinke, S., 2016. Palaeoclimatic insights into forcing and response of monsoon rainfall, *Nature*, 533, 191–199

O'Gorman, P.A. and Muller, C.J., 2010. How closely do changes in surface and column water vapor follow Clausius–Clapeyron scaling in climate change simulations?. *Environmental Research Letters*, 5(2), p.025207.

Seth, A., Giannini, A., Rojas, M., Rauscher, S.A., Bordoni, S., Singh, D. and Camargo, S.J., 2019. Monsoon responses to climate changes—connecting past, present and future. *Current Climate Change Reports*, 5(2), pp.63-79.

Reviewer #2 (Remarks to the Author):

In "New insights into speleothem oxygen isotope deciphered by multi-proxy records and climate model simulations in southern Oman," Tian et al. present new fluid inclusion δD and carbonate $\delta^{17}O$ measurements from south Oman stalagmite Q5, paired with air parcel back trajectory analysis and climate model simulations, to comment on interpretation of the published Q5 $\delta^{18}O_c$ record, particularly with respect to Indian Summer Monsoon strength (and North African summer monsoon strength) captured in southern Oman over the Holocene. The paper concludes that the Q5 $\delta^{18}O_c$ record indicates the change of a super-monsoon regime across the Holocene.

The fluid inclusion δD and carbonate $\delta^{17}O$ measurements are an excellent addition to this significant speleothem $\delta^{18}O$ record (published almost 2 decades ago now) using the latest advances in the field, and the authors' moisture trajectory hypothesis discussion, which is well supported by the new proxy records and model analysis, will be of interest to many in the community interested in African and Asian summer monsoon mechanisms. However, there were several instances in the text in which I found the authors taking too much liberty in brushing over topics without much explanation, assuming too much in my opinion that the reader would take the authors at their word, which I don't really appreciate, particularly in a journal such as *Nature Communications*. I detail these below. I thus propose a major revision before publication in this journal.

Major comments:

1. The methods of the FLEXPART analysis are poorly described, and thus it is unclear to me how 'E-P diagnosed from 12-days back trajectories using FLEXPART' shown in Fig.1 was calculated, or what the figure is even displaying. The figure caption does not state explicitly from which point the back trajectories were back tracked from, or on what dates, nor otherwise refer the reader to the methods section. In the methods FLEXPART section, there is limited information on how the FLEXPART simulation was performed: "We calculated the mean net moisture flux based on E-P and backtracked elements that had a loss ($E-P < 0$) over the Qunf Cave region ($16-18^\circ N$, $52-56^\circ E$) for 12 days." I understand picking out Qunf cave area air parcel elements with an $E-P < 0$ during JJA, and then performing a back trajectory analysis on each of them, but I don't understand the jump from having 1000s(?) of air parcel back trajectories to the map display shown in Fig. 1. This is missing from the methods.
2. $\delta^{17}O$ was not introduced at all- I imagine this proxy is not familiar to most broad audience readers. Providing a list of references for the reader to investigate and learn about on their own is not good enough for a *Nat Comms* article in my opinion.
3. There is no information included in the article (or the supplement) on how relative humidity was calculated from the $\delta^{17}O$ proxy. In the "Triple oxygen isotope analysis" methods section (methods was not referenced in the main article text), there is an equation listed, but no citation with this equation, and no explanation on where it comes from. The sentence prior to the equation says that temperature was calculated from TraCE-21k... is the equation dependent on T? If so, what is its sensitivity to T? TraCE-21k is relatively low resolution, and this site is near the coast. Also, why TraCE-21k when EC-Earth simulations were run specifically for this paper? Further, not all the variables in the equation are explicitly stated. Overall, I found this section of the methods incredibly confusing.
4. Line 187 introduces the terms "East African monsoon" and "West African monsoon" without any explanation... until this point the African moisture has been described as the "North Africa summer monsoon". I'm confused. This is another example of the text using terms without any description.

5. Line 195 states “the proxies of the local hydroclimate conditions are then a critical complement” to the regional $\delta^{18}\text{O}$, δD , $\delta^{17}\text{O}$ proxies. The authors however do not include any explanation in the main text on how their local hydroclimate proxies ($^{234}\text{U}/^{238}\text{U}$, $\delta^{13}\text{C}$, Mg/Ca , Sr/Ca , or Ba/Ca) are interpreted. There is text in the supplement, but in my opinion if the information is ‘critical’, the proxy interpretation should not be 100% exported to the supplement section of the article. Indeed, the interpretation of these proxies is quite complex and interesting in this situation— there is an early-late Holocene pattern in $\delta^{13}\text{C}$, $^{234}\text{U}/^{238}\text{U}$, and Mg/Ca , Sr/Ca , and Ba/Ca , but the different proxies do not complement each other in a simple manner. In my opinion the section on the local hydroclimate proxies needs to either be given an appropriate amount of discussion text in the manuscript (including perhaps the summary paragraph from the supplement text), or removed and published as a separate article.

6. Paragraph beginning Line 214 discusses Clemens et al. work, who have a hypothesis on the mechanisms responsible for ISM strength. The citations listed are very old (Clemens et al., 1991; Clemens and Prell, 2003), which I found a bit disingenuous, particularly as this paper’s senior author’s recent work on monsoon strength mechanisms is well cited in the paper (e.g. Cheng et al., 2022; Cheng et al., 2021). Perhaps in this paragraph the authors could comment on Clemens et al., *Science Advances*, 2021 to acknowledge/address more up-to-date research by this group?

7. In the ^{230}Th Dating methods section, the authors state “All ages are in stratigraphic order within dating uncertainties.” This is a false statement, as demonstrated in Fig. S1. There are several ages from Fleitmann et al. 2003 which appear to be ~ 500 yr offset from the new U-Th ages—though I cannot comment on exact differences because none of the U-Th data is provided in a table. I find it disappointing that the group of authors, which includes Fleitmann as second author, have completely ignored this mismatch, and acted in the manuscript as if it doesn’t exist. A section is needed in the supplement describing why some previously published Q5 U-Th ages have been ignored in the final age model for this study, and if the new Q5 age model modifies any previously published age-sensitive proxy interpretations from Q5 $\delta^{18}\text{O}$ and $\delta^{13}\text{C}$.

Minor comments:

-NHSI, first appearing on Line 38, is not defined. From what latitude, and what date(s) or longitude of perihelion(s), is the insolation calculated?

-Line 141- what is a significant change? Please quantify

-Given that Africa and Asia both start with ‘A’, I would avoid using the monsoon acronyms: NASM, ASM, ISM etc. It makes the text unnecessarily confusing.

-Line 174 – include the period of the -2‰ to 0‰ increase, as it does not continue all the way to the present. Likewise Line 175/176 is false: there is a hiatus at 2 ka, and the $\delta^{18}\text{O}$ values beginning at 1ka decrease through their ~ 500 yrs of growth.

-Fluid inclusions methods section says the reproducibility of crushed speleothem samples is 2‰ , but the results shown in Fig 2 have variable error bars, many much less than 2‰ . Should the methods section say “maximum 2‰ ”?

-Lines 326/327 are confusing. Was the CMIP6 configuration of EC-Earth3-veg-LR used for both the PI and 8k simulation? If so this could be stated more clearly. If not, why not? Is there a difference between the CMIP6-PMIP4 and the CMIP6 protocol?

Reviewer #1 (Remarks to the Author):

General comments

Tian et al. in this manuscript, have attempted to interpret multiple proxies taken from a cave in Oman, in the context of Indian monsoon over the Holocene. They have tried to substantiate and elucidate their results with climate model simulations. They find that the source of moisture changed across the Holocene, due to changes in large scale monsoon circulation. While interesting, I do not find this result to be novel enough to justify publication in a high impact journal and warrants a rejection. I find a bit of ambiguity/lack of clarity in some of the sentences which allows for a benefit of doubt. Hence, I want to give the authors a chance to respond to the following major concerns, and I recommend a major revision instead.

Major concerns

1. Novelty of the work: Battisti et al. 2014, have already established that Qunf cave proxy does not represent precipitation intensity. It merely captures the isotope fractionation of water vapor - implying the source of moisture as the driver of $\delta^{18}\text{O}$. A number of recent literature (eg. Dayem et al. 2010, Krklec et al. 2014, Kathayat et al. 2020; to name a few) have demonstrated that isotope fractionation of oxygen from speleothems are related to changes in the source of moisture. I find the current study one of the many of this growing body of literature on this aspect.

This is a very good question.

The Holocene $\delta^{18}\text{O}_c$ record of stalagmite Q5 is one of the most important records in the Asian summer monsoon/Indian summer monsoon regions, which is rather unique in the Arabian Peninsula till today. The broad scientific significance of this record is attested by more than 1700 citations (*Google Scholar*), since Fleitmann et al. (2003) published it in *Science*. In the past two decades, the fast developments in both proxy and modelling research forefronts have made it possible to further study this unique site with multi-proxy and model-data comparison approaches, which is critical to understand a number of scientific issues in debate.

At present, the interpretation of the climate significance of speleothem $\delta^{18}\text{O}_c$ alone remains tentative. While a set of modelling studies had been carried out to resolve this issue (e.g., Battisti et al., 2014; Bosmans et al., 2018; Hsu et al., 2010; Jaliha et al., 2019; Jaliha et al., 2020; Jaliha et al., 2022), the Holocene evolution of the Indian summer monsoon across the Arabian Peninsula is still ambiguous, manifesting for instance, a so-called “sea-land precession-phase paradox”, particularly for the Holocene (e.g., Cheng et al., 2021, 2022). This is largely because of the lack of proxy-based evidence besides the $\delta^{18}\text{O}_c$ record published two decades ago, thus calling for multi-proxy data from the region in order to make a new in-depth and comprehensive

data-model comparison.

Most of the previous studies up till now focused mainly on analysis of modern moisture sources and hydroclimate modelling (e.g., Battisti et al., 2014; Dayem et al., 2010; Kathayat et al., 2021; Krklec and Dominguez-Villar, 2014). In this study, additional to modern hydroclimate analyses, we provided substantial new proxy-based evidence, including $(^{234}\text{U}/^{238}\text{U})_0$, $\delta^{13}\text{C}$, trace element, $\Delta^{17}\text{O}$ and fluid inclusion dataset, and new modelling results, such as precipitation, 850 hPa wind, and near-surface relative humidity. This allows us not only to reconstruct the Holocene hydroclimate evolution more robustly, but also to accomplish a critical comparison between new multi-proxies and model results for the critical site.

Accordingly, we have made the following changes:

Lines 29-30 in the abstract: “*This study thus provides critical proxy data in support of previous and new model results.*”

Lines 64-66 in the introduction: “*It has been highly cited during the past two decades and remains a unique Holocene $\delta^{18}\text{O}_c$ record in the area up till now.*”

Lines 107-108 in the introduction: “*These questions call for further investigations, especially with multi-proxy data.*”

Lines 309-311 in the conclusion section: “*Our multi-proxy records and new modelling results provide new insights into the interpretation of the Qunf $\delta^{18}\text{O}_c$ record and the Holocene hydroclimate history in the southern Arabian Peninsula.*”

2. Reconstruction of Humidity: The authors have calculated humidity based on $\Delta^{17}\text{O}$ and temperature from a transient simulation TraCE-21k. One should be careful while combining parameters from proxy and a climate model. Moreover, it is not clear, how the temperature was used in their equation. The authors should also show RH calculated entirely from the output of TraCE-21k. Does RH calculated from TraCE-21k match the reconstructed RH?

We agree with the reviewer.

Temperature is used to calculate the $\delta^{18}\text{O}$ fractionation ($^{18}\alpha_{\text{carbonate/water}}$) between paired cave dripwater and the carbonate precipitation from the water by the following equation (Kim and O'Neil, 1997):

$$1000\ln^{18}\alpha_{\text{carbonate/water}} = 18.03(10^3 T^{-1}) - 32.42 \quad (1)$$

We agree with the reviewer's comment that combining parameters from proxy and a climate model might bring in some uncertainties. In the revised manuscript, we used the summer SST from ODP site 723A (Naidu and Malmgren, 2005), instead of the TraCE-21k temperature, to calculate RH. The calculated RH results from the SST and TraCE-21k are similar within the uncertainty (Fig. R1).

Figure R1. Comparison between the relative humidity (RH) results calculated by temperature from the TraCE-21, EC-Earth and summer SST in ODP site 723A. Individual errors are standard errors of the corresponding measurements propagated from $\delta^{17}\text{O}_c$ and $\delta^{18}\text{O}_c$ of speleothem carbonates (Sha et al., 2020; Sha et al., 2021). The triple oxygen isotope compositions of parent water are calculated by the Monte Carlo model (MC=1000) in combination with statistical estimates of uncertainty.

We have added more information and related equations in the revised manuscript and supplementary file to provide the details about the calculation of $\Delta^{17}\text{O}$ and RH.

3. Interpretation based on relative humidity: The authors have shown that RH in the proxy was different in early Holocene compared to the late Holocene. This can be due to either a change in local RH or a change in source of moisture or a combination of these two factors. Since, there is no definite way to isolate these two effects on the proxy, the authors have used EC-Earth simulations. It is a well-known fact that most CMIP models simulate a constant RH (e.g., Gormon & Muller 2010, Allan et al. 2020, Douville et al. 2022). Consistent with this, EC-Earth shows no changes in RH, except over North African summer monsoon region and the Arabian Peninsula.

This is primarily due to the fact that these regions are deserts in the control. The large change in the hydrology during the early Holocene leads to large change in RH, in these regions. The model shows an increase in RH as expected. However, figure 2c suggests that the proxy has a recorded a decrease in RH. The direction of change suggested by the proxy and model are not consistent. This inconsistency has not been discussed. A change in moisture source surely cannot have such a drastic impact. Even if I assume the authors interpretation is correct, Qunf is too close to the Arabian Sea (roughly 15 km) - I'd still say that moisture from the Arabian Sea dominates. In fact, a visual examination of figure 3b suggests that almost entire moisture and winds into the cave site are from the Arabian Sea.

Suggestion: use FLEXPART to identify the source. Or the tagging technique used in Kathayat et al. 2020.

Thanks for the good questions/comments.

(1) In our study, the $\Delta^{17}\text{O}$ -based reconstruction of the RH at the moisture source shows a $\sim 15\%$ increase from the middle to late Holocene (Fig. 2c), which is apparently consistent with our model results for the ultimate ocean sources (Fig. 3a). On the other hand, the simulated RH at the Arabian Sea and Qunf Cave site increases by $\sim 6\%$ from 8K to PI (Figs. 3a and R2). This is consistent with our speleothem proxy records [$(^{234}\text{U}/^{238}\text{U})_0$, $\delta^{13}\text{C}$, and trace element ratios (Mg/Ca, Sr/Ca, Ba/Ca)] that also indicate a drier condition (low P-E) at the Qunf Cave site during the early to middle Holocene compared to the late Holocene. Thus, both model simulation and proxy record show a similar increasing trend from 8K to PI in terms of local hydroclimate conditions at the cave site. Nonetheless, we admit that the model simulation of RH remains a challenge, especially regarding its uncertainties.

In the revised manuscript, we have added the above discussions.

Lines 169-174: *“The model simulations reveal a $\sim 6\%$ increase in relative humidity at the equatorial Atlantic, Arabian Sea and Qunf Cave site from 8K to PI (Fig. 3a). The $\Delta^{17}\text{O}$ -based reconstruction of relative humidity at the moisture source shows a $\sim 15\%$ increase from middle to late Holocene (Fig. 2c). The direction of these Holocene trends is consistent. Taking into account the uncertainties of the modelled and reconstructed relative humidity, these datasets show strong similarities.”*

Lines 249-252: *“Both model simulation and proxy records show a similar increasing trend in local hydroclimate conditions at the cave site from 8K to PI, the early to middle Holocene was drier (lower P-E) during early to middle Holocene compared to the late Holocene (See supplementary text).”*

Figure R2. Model result of 8K-PI near-surface relative humidity (% in colour-scale). Also shown is the EC-Earth grid cell (green box) containing the Qunf Cave site (red star).

(2) Currently, the FLEXPART cannot provide the analysis on moisture back trajectory. Since the EC-Earth has not yet implemented the water tagging diagnostics

in its simulations and output, we instead use the iCESM from Hu et al. (2019) for the analysis, which is similar to that used in Kathayat et al. (2021). The iCESM tagging results show that besides the Arabian Sea as the dominant moisture source for the Qunf Cave site, North Africa and the Arabian Peninsula contribute a large percentage of moisture to the Qunf Cave site (Fig. R3). Therefore, we suggest that when NHSI was high during the early and middle Holocene, the North African monsoon may likely contribute more water vapor to the Qunf Cave site.

Figure R3 has been added to the supplementary material as Supplementary Fig. 1.

We have added the above information in the revised main text (lines 123-126):

“Meanwhile, the moisture tagging analysis and simulations show that the African continent, Mediterranean and Iranian Plateau also provide considerable moisture to the Qunf Cave site (Fig. 1), particularly from North Africa and the Arabian Peninsula during summer (JJA) (Supplementary Fig. 1).”

Figure R3. Seasonal moisture contribution from each source to the Qunf Cave site in the iCESM simulation (1954-2012).

(3) The wind arrows in Fig. 3b show a strong westerly-southwesterly wind anomaly from the African rain belt towards the Arabian Peninsula and Oman. A number of previous model studies (e.g., Patricola and Cook, 2007; Battisti et al., 2014; Hu et al., 2019) also suggest that during the high NHSI periods, the moisture transformed from the tropical Atlantic across Northeast Africa as far as East Africa and the Arabian Peninsula appears to increase. The study from Hoti Cave speleothem, Northeast Oman, also suggests that “*greatly enhanced rainfall was caused by an intensification and greater northward extension of the African and Indian monsoons into Arabia*” (Fleitmann et al., 2022). Overall, the Arabian Sea is likely the dominant moisture source during the Holocene. In the meantime, moisture originating from the North African summer monsoon may also contribute to the precipitation at the cave site, although it is difficult to precisely assess the percentage of the contribution of each moisture source. However, we suspect that the contribution of the moisture from the North African

monsoon in the middle Holocene would be higher than or at least comparable to today's percentage inferred by the moisture tagging simulations (Figure R3).

4. Use of FLEXPART The authors have used FLEXPART in the modern context only. They have identified moisture sources for precipitation at the site of the cave based on modern data.

They have used two yrs - one with lowest and another highest annual precipitation in modern climate. For more robustness I recommend that the authors instead use a composite of high and low rainfall years (defined by precip greater than or lower than 1 sigma for example). As acknowledged by the authors, the sources of moisture change across the Holocene. Still, the authors use results based on modern day FLEXPART analysis to interpret their proxy. I suggest that the authors do the analysis with the 8ka and PI simulations to make a strong case.

This is a good suggestion.

We have rerun the FLEXPART simulations based on ERA-interim precipitation (1979-2019) and identified the wet and dry composites from the three driest and three wettest ($>1\sigma$) June-August (JJA) years in our study area (17°N - 17.75°N , 54°E - 54.75°E), respectively. The results show that the African Continent, Mediterranean and Iranian Plateau also contribute considerable moisture to the Qunf Cave site, consistent with our previous result (Fig. 1).

Figure 1. Qunf Cave (yellow asterisk) JJA evaporation minus precipitation (E-P) diagnosed from 12-days back trajectories in June-August using the FLEXPART for (a) highest precipitation years (1983, 1986 and 1996) and (b) lowest precipitation years (1999, 2000 and 2012). Positive values (red) indicate a net moisture supply, while negative values (blue) indicate water vapor condensation and precipitation. Sites of other caves (yellow circles) and marine sediment cores (green boxes) discussed in the text that contain important Holocene paleoclimate records (discussed in the text) are also shown: HT, Hoti Cave; JT, Jeita Cave; SQ, Soreq Cave; TM, Tianmen Cave; SH, Sahiya Cave; ML, Mawmluh.

In general, the FLEXPART simulations do not provide direct explanations for our proxies, neither for PI scenarios, but rather for setting a baseline for the current

atmospheric circulation and moisture sources to the Qunf Cave site. That is, the FLEXPART cannot perform back trajectory analysis for 8K and PI. Instead, however, our EC-Earth simulations show precipitation and wind differences between 8K and PI, suggesting a stronger southwesterly wind strength and more precipitation over North Africa during 8K compared with PI. Besides, speleothem fluid inclusion δD and triple oxygen isotope data provide a line of evidence that the North African summer monsoon may possibly contribute more precipitation at the Quaf Cave site during the early to middle Holocene, although the Indian Ocean remains an important source.

5. Insufficient literature Line 85-88 and Lines 221 - 225: I have been following the recent developments in the debate between marine and terrestrial proxies of monsoons quite closely. This debate has been largely resolved recently. Therefore, it is important for the benefit of the readers to have a couple of additional sentences. This will help the readers get a perspective of why it is important to isolate processes in the Arabian Sea and those related to Qunf - which is quite essential for this paper. The following papers must be included: Hsu et al 2010 was the first to suggest the opposing response of land and ocean monsoons, followed by Battisti et al. 2014, Bosmans et al. 2018. Jalihal et al. 2019 & Jalihal et al. 2020, were the first papers to connect it to the opposing marine and terrestrial proxy response, and provide a mechanism for the opposite response. Later, Cheng et al, 2021 coined the term “land-sea precession phase paradox”.

We thank the reviewer for his/her good suggestion. Accordingly, we have added the above content suggested by the reviewer in the revised introduction and discussion sections.

Lines 90-94: *“Hsu et al. (2010) were the first to show that their climate model actually reproduced this opposite response between land and ocean to NHSI at the precession band. This study was followed by multiple climate model simulations that confirmed the results from Hsu et al. (Zhang et al., 2021; Battisti et al., 2014; Bosmans et al., 2018; Jalihal et al., 2019, 2020 and 2022). Later, Cheng et al. (2021) coined the term ‘land-sea precession phase paradox’ in the research forefront of the Asian summer monsoon, referring to the contrast between speleothem and marine proxy records.”*

Lines 96-98: *“Recently, Jalihal et al.(2019,2020) explained the mechanisms for the opposing marine and terrestrial proxy response in the Indian summer monsoon domain.”*

Lines 288-294: *“Model simulations reveal contrasting hydroclimatic responses between land and ocean to NHSI changes at the precession band (Hsu et al., 2010; Battisti et al., 2014; Bosmans et al., 2018). This phenomenon has been referred to as the “land-sea precession phase paradox” (Cheng et al., 2022). In the ISM regions, Jalihal et al. (2019, 2020, 2022) investigated the opposing marine and terrestrial responses, both empirically and theoretically. They discovered that, in addition to NHSI, precipitation changes over the ocean are influenced by changes in surface energy fluxes and vertical stability.”*

Minor points

1. Obsolete theory of monsoon: Line 69-70: The authors refer to the enhancing land-sea thermal contrast as major cause of stronger monsoon. Over the last couple of decades, studies have discredited the land-sea thermal contrast theory of monsoons and moist static energy-based theories have gained momentum (see review papers eg. Gadgil 2018, Biasutti et al. 2018, Hill 2020). This has been used to understand orbital-scale monsoon variability (D’Agostino et al. 2019, 2020, Jalihal et al 2019, 2020). Accordingly text needs to be modified at appropriate places eg. lines 210-213.

This is indeed a good suggestion.

We agree with the reviewer. For over 300 years (Halley, 1686), the monsoon has been considered to be a gigantic land–sea breeze. The traditional conception of monsoons as land-sea breezes have been increasingly challenged by the modern perspective of monsoons as an integral component of the global atmospheric circulation and hydroclimate (e.g., Gadgil et al., 2018; Hill et al., 2019). Of note are three primary theoretical conceptions of monsoons: one based on convective quasi-equilibrium (CQE) (e.g., Emanuel et al., 1994; Nie et al., 2010), another founded on the moist static energy (MSE) budget (e.g., Neelin et al., 1987; Chou et al., 2001), and one that frames the monsoon as an extension of the zonal-mean ITCZ (e.g., Biasutti et al., 2018; Hill 2019). While those theoretical frameworks can better explain some important aspects of the modern monsoon variability, especially regarding monsoon rainfalls and thermodynamics, the interpretations of paleoclimate variations on a wide range of timescales are still in development (e.g., D’Agostino et al. 2019, 2020; Jalihal et al., 2019, 2020). Physically, the summer barometric differentials and other boundary conditions in a vast spatial-scale, regardless of the cause of it, would drive the Asian summer monsoon circulation (e.g., at 850 hPa) (Fig. R4) and associated large-scale vapor flux (Fig. R5). In terms of precipitation $\delta^{18}\text{O}$ at the cave site, prior to the onset of the Asian summer monsoon, the moisture source is rather close and thus precipitation is characterized by heavier $\delta^{18}\text{O}$. After the summer monsoon onset, the spatial-scale of monsoon circulation and moisture from remote sources increase dramatically (Figs. R4 and R5), both resulting in the lighter precipitation $\delta^{18}\text{O}$ at cave site. As such, cave $\delta^{18}\text{O}$ in principle, indicates mainly the dynamic aspect or the monsoon circulation, rather than a direct proxy of local rainfall amount or thermodynamics that are shown via other proxies [e.g., trace elements, $(^{234}\text{U}/^{238}\text{U})_0$, and $\delta^{13}\text{C}$].

Figure R4 and Figure R5 have been added to the supplementary material as Supplementary Fig. 8 and Supplementary Fig. 9.

Accordingly, we have expanded our main text to reflect the development in monsoon theory:

Lines 76-78: *“However, some model simulations reveal distinct regional divergences of the spatial patterns of rainfall and wind across the precession cycle in the Asian summer monsoon domain.”*

Lines 256-273: *“The understanding of monsoons as an integral component of the global atmospheric circulation and hydroclimate is becoming prevalent (e.g., Gadgil et*

al., 2018; Hill et al., 2019). Of note are three primary theoretical concepts of monsoons: one based on convective quasi-equilibrium (CQE) (e.g., Emanuel et al., 1994; Nie et al., 2010), another focused on the moist static energy (MSE) budget (e.g., Neelin et al., 1987; Chou et al., 2001), and a third that frames the monsoon as an extension of the zonal-mean ITCZ (e.g., Biasutti et al., 2018; Hill 2019). While these theoretical frameworks help explain certain aspects of modern monsoon variability, especially regarding monsoon rainfalls and thermodynamics, the interpretations of paleoclimate variations across various timescales are still in development (e.g., Jalihal et al., 2019, 2020; D'Agostino et al. 2019, 2020).

Physically, the summer barometric differentials and other boundary conditions on a large spatial-scale, regardless of their cause, drive the Asian summer monsoon circulation (e.g., at 850 hPa) (Supplementary Fig. 8) and the associated large-scale vapor flux (Supplementary Fig. 9). In terms of precipitation $\delta^{18}O_p$ at the cave site, prior to the onset of the Asian summer monsoon, the moisture source is relatively, resulting in heavier $\delta^{18}O_p$. After the summer monsoon onset, the spatial-scale of monsoon circulation and moisture from remote sources increase dramatically (Supplementary Figs. 8 and 9), both leading to lighter precipitation $\delta^{18}O_p$ at the cave site. Consequently, cave $\delta^{18}O_c$ mainly indicates the dynamic aspect of the monsoon circulation, with a small portion of the variability explained by local rainfall amount or thermodynamics. Local rainfall amount or P-E is, however, better reflected by other proxies [e.g., trace elements, $(^{234}U/^{238}U)_0$, and $\delta^{13}C$].”

Fig. R4. 850 hPa wind of (a) pre-monsoon season (March to May, MAM), and (b) monsoon season (June to July) during 2018 to 2022 from ERA5.

Fig. R5. Vertically integrated vapor flux of (a) pre-monsoon season (March 15 to May 15), and (b) monsoon season (June 1 to July 15) during 1981 to 2010 from the Nation Climate Centre of China.

2. Line 35: Include citations to Mohtadi et al. 2014, Seth et al. 2019

The two references have been cited in line 36.

3. Line 53: As noted later in the literature, it is not just the fractionations integrated from tropical oceans to cave site, but mixing from other moisture sources etc. It will be good to mention these factors in the introduction itself.

We have revised the text to mention these factors in the introduction (lines 54-57):

“Theoretically, speleothem $\delta^{18}O_c$, as a proxy of the $\delta^{18}O_p$ at the cave site, is related to oxygen-isotope fractionations integrated from tropical oceans to the cave site along the moisture transport trajectory. Besides, the influence of other factors on speleothem $\delta^{18}O_c$, such as mixing of different moisture sources and rainfall amount, should not be dismissed.”

4. Line 101: ISMt -> ISM

All acronyms for monsoons have been changed to full spelling in the revised manuscript.

5. Line 166-168: How is this statement different from the main conclusion of Battisti et al. 2014? Perhaps, a more correct statement would be that the authors have an observational evidence for a result proposed by Battisti et al. based on models.

Good suggestion.

Indeed, in this study we present multi-proxy evidence from Qunf Cave, a set of model simulations on the Holocene hydroclimate evolution focused in the region, and thus a close comparison between the observations and Holocene hydroclimate condition model results. Our study provides an ideal complement to Battisti et al. (2014), which focus on the $\delta^{18}O_c$ interpretation in the pan-Asian climatic system based on the

“high-insolation” (218 ka BP) and “low-insolation” (207 ka BP) experiments, the high and low extrema in JJA insolation over the past 950,000 years, especially compared to the Holocene. Regarding the $\delta^{18}\text{O}_c$ interpretation and the moisture contribution from the tropical Atlantic via northeast Africa to the Arabian Peninsula, our results are consistent with the conclusion of Battisti et al. (2014) based on the experiments of “high and low extrema”.

We have revised the statement (lines 189-196) as:

“The Qunf multi-proxy dataset, combined with a set of model simulations on the Holocene hydroclimate evolution in the region, allow for a thorough comparison between observed and modelled Holocene hydroclimate conditions. We propose that, in addition to the water vapor transported by the Indian summer monsoon from the adjacent Arabian Sea, the North African summer monsoon may have contributed remote moisture from the tropical Atlantic via northeast Africa to Qunf Cave during the early and middle Holocene. This finding offers robust evidence supporting earlier model results (Battisti et al., 2014; Bosmans et al., 2018)”

6. Line 174-175: “Currently, the ISM is prevailing along the southern Arabian Peninsula”. This statement is wrong. How can Indian summer monsoon be prevalent over the Arabian peninsula?

I think the authors mean, the effect of ISM is prevalent over the Arabian peninsula.

We agree with the reviewer. This statement (lines 202-203) has been revised to:

“Currently, the effect of the Indian summer monsoon predominates over the southern Arabian Peninsula.”

References

- Bar-Matthews, M., Ayalon, A., Matthews, A., Sass, E., Halicz, L., 1996. Carbon and oxygen isotope study of the active water-carbonate system in a karstic Mediterranean cave: Implications for paleoclimate research in semiarid regions. *Geochimica Et Cosmochimica Acta* 60, 235-241.
- Battisti, D.S., Ding, Q., Roe, G.H., 2014. Coherent pan-Asian climatic and isotopic response to orbital forcing of tropical insolation. *Journal of Geophysical Research: Atmospheres* 119, 11,997-912,020.
- Biasutti, M., Voigt, A., Boos, W.R., Braconnot, P., Hargreaves, J.C., Harrison, S.P., Kang, S.M., Mapes, B.E., Scheff, J., Schumacher, C., Sobel, A.H., Xie, S.P., 2018. Global energetics and local physics as drivers of past, present and future monsoons. *Nature Geoscience* 11, 392-+.
- Bosmans, J.H.C., Erb, M.P., Dolan, A.M., Drijfhout, S.S., Tuenter, E., Hilgen, F.J., Edge, D., Pope, J.O., Lourens, L.J., 2018. Response of the Asian summer monsoons to idealized precession and obliquity forcing in a set of GCMs. *Quaternary Science Reviews* 188, 121-135.
- Cheng, H. et al., 2022. Milankovitch theory and monsoon. *The Innovation* 3, 100338.
- Cheng, H., Zhang, H., Cai, Y., Shi, Z., Yi, L., Deng, C., Hao, Q., Peng, Y., Sinha, A., Li, H., Zhao, J., Tian, Y., Baker, J., Perez-Mejías, C., 2021. Orbital-scale Asian summer monsoon variations:

- Paradox and exploration. *Science China Earth Sciences* 64, 529-544.
- Chou, C., Neelin, J. D., & Su, H., 2001. Ocean-atmosphere-land feedbacks in an idealized monsoon. *Quarterly Journal of the Royal Meteorological Society*, 127(576), 1869-1891.
- D'Agostino, R. et al., 2020. Contrasting Southern Hemisphere Monsoon Response: MidHolocene Orbital Forcing versus Future Greenhouse Gas-Induced Global Warming. *J. Clim.* **33**, 9595-9613, doi:10.1175/Jcli-D-19-0672.1.
- D'Agostino, R., Bader, J., Sordani, S., Ferreira, D. & Jungclaus, J., 2019. Northern Hemisphere Monsoon Response to Mid-Holocene Orbital Forcing and Greenhouse Gas-Induced Global Warming. *Geophys. Res. Lett.* **46**, 1591-1601, doi:10.1029/2018gl081589.
- Dayem, K.E., Molnar, P., Battisti, D.S., Roe, G.H., 2010. Lessons learned from oxygen isotopes in modern precipitation applied to interpretation of speleothem records of paleoclimate from eastern Asia. *Earth and Planetary Science Letters* 295, 219-230.
- Emanuel, K. A., David Neelin, J., & Bretherton, C. S., 1994. On large-scale circulations in convecting atmospheres. *Quarterly Journal of the Royal Meteorological Society*, 120(519), 1111-1143.
- Gadgil, S., 2018. The monsoon system: Land–sea breeze or the ITCZ? *Journal of Earth System Science*, 127, 1-29.
- Halley, R., 1686. An historical account of the trade winds and monsoons, etc. *Phil. Trans. Roy. Soc. London*, 16,153-158.
- Hill, S.A., 2019. Theories for Past and Future Monsoon Rainfall Changes. *Curr Clim Change Rep* 5, 160-171.
- Hsu, Y.H., Chou, C., Wei, K.Y., 2010. Land-Ocean Asymmetry of Tropical Precipitation Changes in the Mid-Holocene. *Journal of Climate* 23, 4133-4151.
- Hu, J., Emile-Geay, J., Tabor, C., Nusbaumer, J., Partin, J., 2019. Deciphering Oxygen Isotope Records From Chinese Speleothems With an Isotope-Enabled Climate Model. *Paleoceanography and Paleoclimatology* 34, 2098-2112.
- Jalihal, C., Bosmans, J.H.C., Srinivasan, J., Chakraborty, A., 2019. The response of tropical precipitation to Earth's precession: the role of energy fluxes and vertical stability. *Climate of the Past* 15, 449-462.
- Jalihal, C., Srinivasan, J., Chakraborty, A., 2020. Different precipitation response over land and ocean to orbital and greenhouse gas forcing. *Scientific Reports* 10.
- Jalihal, C., Srinivasan, J., Chakraborty, A., 2022. Response of the Low-Level Jet to Precession and Its Implications for Proxies of the Indian Monsoon. *Geophysical Research Letters* 49.
- Kathayat, G., Sinha, A., Tanoue, M., Yoshimura, K., Li, H., Zhang, H., Cheng, H., 2021. Interannual oxygen isotope variability in Indian summer monsoon precipitation reflects changes in moisture sources. *Communications Earth & Environment* 2, 1-10.
- Kim, S.-T., O'Neil, J.R., 1997. Equilibrium and nonequilibrium oxygen isotope effects in synthetic carbonates. *Geochimica et Cosmochimica Acta* 61, 3461-3475.
- Krklec, K., Dominguez-Villar, D., 2014. Quantification of the impact of moisture source regions on the oxygen isotope composition of precipitation over Eagle Cave, central Spain. *Geochimica et cosmochimica acta* 134, 39-54.
- Naidu, P.D., Malmgren, B.A., 2005. Seasonal sea surface temperature contrast between the Holocene and last glacial period in the western Arabian Sea (Ocean Drilling Project Site 723A): Modulated by monsoon upwelling. *Paleoceanography* 20.
- Neelin, J. D., & Held, I. M., 1987. Modeling tropical convergence based on the moist static energy budget.

- Monthly Weather Review, 115(1), 3-12.
- Nie, J., Boos, W. R., & Kuang, Z., 2010. Observational evaluation of a convective quasi-equilibrium view of monsoons. *Journal of Climate*, 23(16), 4416-4428.
- Patricola, C.M., Cook, K.H., 2007. Dynamics of the West African monsoon under mid-Holocene precessional forcing: Regional climate model simulations. *Journal of Climate* 20, 694-716.
- Sha, L.J., Maha, S., Duan, P.Z., Luz, B., Zhang, P., Baker, J., Zong, B.Y., Ning, Y.F., Brahim, Y.A., Zhang, H.W., Edwards, R.L., Cheng, H., 2020. A novel application of triple oxygen isotope ratios of speleothems. *Geochimica Et Cosmochimica Acta* 270, 360-378.
- Sha, L.J., Mahata, S., Duan, P.Z., Zong, B.Y., Ning, Y.F., Zhang, P., Wang, J., Cai, Y.J., Cheng, H., 2021. Preparation of high-precision CO₂ with known triple oxygen isotope for oxygen isotope analysis. *Isot Environ Healt S* 57, 443-456.

REVIEWER COMMENTS

Reviewer #1 (Remarks to the Author):

General comments:

Overall, I'm satisfied with the response by the authors, although some do not entirely convince me. The manuscript has also been modified. However, the text needs some minor corrections. Furthermore, I would have liked to see iCESM tagging analysis for the mid-Holocene or early Holocene to substantiate their results. I do not, however, consider this necessary. Hence, I recommend publication only after the authors have addressed the following.

B) Concerns from the author's response:

1. Reply 1: The authors have said that they provide observational evidence for previous model-based results. This implies the insights from speleothem are not "New," and a change in title might be necessary.

2. Reply 2: This is a remarkable result. The match between core & TraCE is good. But which simulations are the EC-Earth? So far, the authors have only described 8ka experiment from EC-Earth. Why is it not shown in the plot?

3. Reply 3: The authors have claimed that, in the modern climate, other than the Arabian Sea, North Africa, and the Arabian peninsula contribute substantially to the moisture over the Qunf cave. Both these regions are dry in the modern climate. How is then the moisture sourced from these regions? Besides, the authors have not mentioned anywhere in the text the contribution from more local sources such as the Red Sea and the Persian Gulf. Contribution from these regions is relatively high, as can be seen from their figures (Fig. 1, and supplementary Fig. 1). In fact, it can be argued that at 8K, the stronger winds from African monsoon pick up moisture from the Red Sea and transport it over to the Qunf cave.

4. Reply 4: The authors have now only used the three highest and lowest precipitation years during the time period (1979-2019). The authors should make a composite of all the years that have precip greater than and less than 1σ .

C) Concerns from the manuscript and supplementary:

1. Line 76-78: The authors should instead re-write the sentence saying that the land-sea contrast theory is not the correct explanation. The current sentence is confusing.

2. Line 138-140: Based on data from northern and southern Oman, the authors suggest that a large-scale re-organization of atmospheric circulation must have taken place. The two cites are too close together to make such a claim. In the later part of the manuscript, the authors have used multiple proxies (Fig. 4). These proxies are spread over sufficiently large region to speculate changes in large-scale circulation. And so this point can be made later in the text.

3. Line 244-246: The authors suggest that increased evaporation at 8ka promotes PCP and a higher trace elements ratio. However, they have substantiated this with a plot of P-E. Wouldn't it be better to show a spatial plot of an anomaly in evaporation instead of P-E?

4. Line 250 contradicts Line 229: In Line 229, authors conclude that precipitation was higher at 8ka, whereas, in Line 250, they say that the early Holocene was drier. In Line 250, I guess they are referring to higher evaporation. However, higher evaporation does not imply a drier climate.

5. Line 275-277: I do not agree with this statement. Precipitation on orbital scales is driven by local insolation (Q. Wen et al., 2022). It is the insolation over the monsoon domains that's important and not the gradient in insolation or higher latitude insolation. However, there is still some ambiguity in

the literature, and the authors need not change this statement if they do not wish to. It would be good to clarify the definition of NHSI in this context.

6. Line 294-305: I find these lines repetitive. For brevity and for the benefit of the reader, the authors must remove these lines as they have already been discussed in the introduction.

7. Line 678: Are the changes in Precip over the Qunf cave significant?

8. Supplementary material Line 101-103: The authors mention that precipitation is notably higher at 8ka over the region of the cave. However, in Fig. 3b, it appears to have only marginally higher precipitation.

9. Supplementary fig 8: The authors have used four years to make a climatology plot. They should use at least 30 yrs, as done in supplementary figure 9.

D) Miscellaneous:

1. Line 23: Replace "climate meaning" with a more appropriate word.

2. Line 25-26: It is not Indian summer monsoon moisture that prevails over southern Oman. In this context, it is better to say moisture from the Arabian Sea prevails in southern Oman.

3. Line 164: Fig 3b is referred to first. Perhaps, it is better if the author's interchange figures 3a and 3b for maintaining flow.

4. Line 322: "between from"

5. Line 323: "a lower P-E. but"

Reviewer #2 (Remarks to the Author):

The authors have comprehensively responded to two detailed reviews, and I am satisfied with the authors' modifications in response to my comments (reviewer 2). I can now recommend this manuscript for publication following a few minor comments regarding the manuscript figures:

1. The top blue/yellow banner in Figure 2 needs to be described in the figure caption. It would also be helpful for the Holocene Humid Period grey bar to be identified in the figure caption.

2. Please add the name of the model in the Figure 3 caption title, instead of just "Simulation results"

Reviewer #1

A) General comments:

Overall, I'm satisfied with the response by the authors, although some do not entirely convince me. The manuscript has also been modified. However, the text needs some minor corrections. Furthermore, I would have liked to see iCESM tagging analysis for the mid-Holocene or early Holocene to substantiate their results. I do not, however, consider this necessary. Hence, I recommend publication only after the authors have addressed the following.

We are glad that you are overall satisfied with our response. Here are our responses to your remaining concerns and suggestions.

B) Concerns/suggestions from the author's response:

1) Reply 1: The authors have said that they provide observational evidence for previous model-based results. This implies the insights from speleothem are not "New," and a change in title might be necessary.

Thank you for the suggestion. We have changed the title to "***Holocene climate change in southern Oman deciphered by speleothem multi-proxy records and climate model simulations***".

2) Reply 2: This is a remarkable result. The match between core & TraCE is good. But which simulations are the EC-Earth? So far, the authors have only described 8ka experiment from EC-Earth. Why is it not shown in the plot?

The RH result calculated by the EC-Earth (blue line in Fig. R1) is very similar to the result calculated by the core and thus covered by the latter (the red line in Fig. R1) in the previous version of the figure. We have moved the blue line to the top so that it is visible. Additionally, our EC-Earth simulation does not have the result before 8 ka BP (Fig. R1).

Figure R1. Comparison between the relative humidity (RH) results calculated by temperature from the TraCE-21, EC-Earth and summer SST from ODP site 723A. Individual errors are standard errors of the corresponding measurements propagated from $\delta^{17}\text{O}_c$ and $\delta^{18}\text{O}_c$ of speleothem carbonates (Sha et al., 2020, 2021). The triple oxygen isotope compositions of parent water are calculated by the Monte Carlo model (MC=1000) in combination with statistical estimates of uncertainty.

3) Reply 3: The authors have claimed that, in the modern climate, other than the Arabian Sea, North Africa, and the Arabian Peninsula contribute substantially to the moisture over the Qunf cave. Both these regions are dry in the modern climate. How is then the moisture sourced from these regions? Besides, the authors have not mentioned anywhere in the text the contribution from more local sources such as the Red Sea and the Persian Gulf. Contribution from these regions is relatively high, as can be seen from their figures (Fig. 1, and supplementary Fig. 1). In fact, it can be argued that at 8K, the stronger winds from African monsoon pick up moisture from the Red Sea and transport it over to the Qunf cave.

This is indeed a good suggestion/comment.

The statement about the moisture contribution from North Africa and the Arabian Peninsula is based on the iCESM simulation results, in which the two regions are set as a whole large tagging region. Meanwhile, the FLEXPART results show more detailed information: Besides the Somali jet, the Red Sea, the Persian Gulf, the Mediterranean and Iranian Plateau also provide considerable moisture to the Qunf Cave site (Fig. 1 and Supplementary Fig. 1). Compared to dry years, wet years have more moisture contributions from the local/nearby regions, including the Red Sea and the Persian Gulf, which is associated with less precipitation over South Asia (Figs. R2 and 1).

We revised the statement (lines 122-126) to: “*In comparison with the drier years, the wetter years in the area have more local moisture contributions, accompanied with less precipitation over South Asia. Meanwhile, the moisture tagging analysis and simulations also indicate that the Red Sea, the Persian Gulf, the Mediterranean and Iranian Plateau provide considerable moisture to the*

Qunf Cave site (Fig. 1 and Supplementary Fig. 1)."

It is sound that the Red Sea contributed moisture to Qunf Cave through the wind of the North African summer monsoon. However, it is difficult to calculate the individual moisture contribution from the North Atlantic, the North African Continent and the Red Sea, respectively. As such, we merely state that the North African summer monsoon brought moisture from the west to the cave site (including the sources of the North Atlantic, the North African Continent and the Red Sea).

4) Reply 4: The authors have now only used the three highest and lowest precipitation years during the time period (1979-2019). The authors should make a composite of all the years that have precip greater than and less than 1σ .

Figure 1 shows the results with the three highest and lowest precipitation years (beyond $\pm 1.5\sigma$), respectively and their difference. In comparison, we provide the results for $\pm 1\sigma$ precipitation anomaly scenarios: -1σ (1999, 2000, 2001, 2012, and 2017), and $+1\sigma$ in (1983, 1986, 1988, 1996, and 2007) in Figure R2, which is broadly similar to the $\pm 1.5\sigma$ scenarios (Fig. 1).

The $\pm 1\sigma$ and $\pm 1.5\sigma$ scenarios denote the outliers outside 68 % and 86.64 % distribution, respectively. The latter ($\pm 1.5\sigma$) in essence shows more extreme cases of stronger-weaker monsoon or wetter-drier variability over southern Oman and surrounding areas. In that regard, we would like to keep the $\pm 1.5\sigma$ results in the manuscript (Fig. 1) and add a plot of their difference (Fig. 1c) as well.

Lines 120-126 are also revised to "*The Somali jet brings considerable moisture to the Qunf Cave site during the summer, mainly from the tropical South Indian Ocean across the equator to the western Arabian Sea (Fig. 1). In comparison with the drier years, the wetter years in the area have more local moisture contributions, accompanied with less precipitation over South Asia. Meanwhile, the moisture tagging analysis and simulations also indicate that the Red Sea, the Persian Gulf, the Mediterranean and Iranian Plateau provide considerable moisture to the Qunf Cave site (Fig. 1 and Supplementary Fig. 1).*"

Figure 1. Qunf Cave (yellow asterisk) JJA evaporation minus precipitation (E-P) diagnosed from 12-day back trajectories in June-August using the FLEXPART. (a) Results for the highest precipitation ($> +1.5\sigma$) years (1983, 1986 and 1996). (b) Results for the lowest precipitation ($< -1.5\sigma$) years (1999, 2000 and 2012). (c) The difference between wet and dry years. Positive values (red) indicate a larger net moisture supply, while negative values (blue) indicate higher water vapor condensation/precipitation. The yellow star depicts Qunf Cave (QF). Sites of other caves (yellow circles) and marine sediment cores (green boxes) discussed in the text that contain important Holocene paleoclimate records (discussed in the text) are also shown: HT, Hoti Cave; JT, Jeita Cave; SQ, Soreq Cave; TM, Tianmen Cave; SH, Sahiya Cave; ML, Mawmluh.

Figure R2. Qunf Cave (yellow asterisk) JJA evaporation minus precipitation (E-P) diagnosed from 12-day back trajectories in June-August using the FLEXPART. (a) Results for the highest precipitation ($> +1\sigma$) years (1983, 1986, 1988, 1996 and 2007). (b) Results for the lowest precipitation ($< -1\sigma$) years (1999, 2000, 2001, 2012 and 2017). (c) The difference between wet and dry years. Positive values (red) indicate a larger net moisture supply, while negative values (blue) indicate higher water vapor condensation/precipitation. The yellow star depicts Qunf Cave.

C) Concerns/suggestions from the manuscript and supplementary:

1) Line 76-78: The authors should instead re-write the sentence saying that the land-sea contrast theory is not the correct explanation. The current sentence is confusing.

We agree with the reviewer. We have rewritten the sentence (lines 74-77) as: “*This interpretation seems to be consistent with the conventional wisdom that increased NHSI enhances the land-sea thermal contrast and, thus, the summer monsoon intensity. However, this classical notion of monsoon has been challenged by a number of recent climate modelling and reanalysis studies.*”.

2) Line 138-140: Based on data from northern and southern Oman, the authors suggest that a large-scale re-organization of atmospheric circulation must have taken place. The two sites are too close together to make such a claim. In the later part of the manuscript, the authors have used multiple proxies (Fig. 4). These proxies are spread over sufficiently large region to speculate changes in

large-scale circulation. And so this point can be made later in the text.

Excellent suggestion. We have revised the sentence (lines 138-139) as: “*Consistently, the δD_f data from Hoti Cave (northern Oman) also show a significant change in the moisture source, seasonality, and amount of rainfall above the cave around 6 ka BP.*”, and moved the statement about the large-scale circulation to the later part of the section (lines 190-193): “*We propose that there was a large-scale reorganization of atmospheric circulation in the region. In addition to the water vapor transported by the Indian summer monsoon from the adjacent Arabian Sea, the North African summer monsoon may have contributed remote moisture from the tropical Atlantic via Northeast Africa to Qunf Cave during the early and middle Holocene.*”

3) Line 244-246: The authors suggest that increased evaporation at 8ka promotes PCP and a higher trace elements ratio. However, they have substantiated this with a plot of P-E. Wouldn't it be better to show a spatial plot of an anomaly in evaporation instead of P-E?

The spatial pattern of anomaly in JJA evaporation between 8K and PI is shown in Figure R3. Both evaporation and precipitation were slightly higher at 8K compared with PI. It appears to be better to display the difference in their P-E, since it in theory indicates the effective wetness (P-E), more relevant to the hydroclimate changes discussed in the section.

Figure R3. EC-Earth simulation of evaporation difference between 8K and PI.

4) Line 250 contradicts Line 229: In Line 229, authors conclude that precipitation was higher at 8ka, whereas, in Line 250, they say that the early Holocene was drier. In Line 250, I guess they are referring to higher evaporation. However, higher evaporation does not imply a drier climate.

We suggest that the precipitation (P) – evaporation (E) (P-E) was lower in the early and middle

Holocene: although both P and E were slightly higher in the time, the overall P-E was lower, implying that the effective wetness was likely lower, thus causing a relatively dry condition.

We have revised the sentence in lines 249-250 to: “*the precipitation-evaporation (P-E) was slightly lower during the early to middle Holocene compared to the late Holocene.*”

5) Line 275-277: I do not agree with this statement. Precipitation on orbital scales is driven by local insolation (Q. Wen et al., 2022). It is the insolation over the monsoon domains that’s important and not the gradient in insolation or higher latitude insolation. However, there is still some ambiguity in the literature, and the authors need not change this statement if they do not wish to. It would be good to clarify the definition of NHSI in this context.

As Beck et al. (2018) mentioned: “*65°N insolation exhibits nearly identical phase, net-range, and pattern of variations as those of the low-latitude (30°N to 30°S) June solar insolation gradient.*”

Particularly, the Indian summer monsoon originates from the Southern Hemisphere near the Mascarene High and transports moisture across the equator, as far as North India. Thus, the interhemispheric differential in tropical insolation (or Summer Inter-Tropical Insolation Gradient, SITIG) may be essential to describe the summer monsoon system. This is because the insolation difference involves both “pull” and “push” forcings from two hemispheres that both propel monsoon changes (e.g., Cai et al., 2006; Cheng et al., 2022; Rohling et al., 2009, and references therein).

We have revised the sentence (lines 273-278) to: “*As the Indian summer monsoon is an interhemispheric monsoon system, the interhemispheric differential in tropical insolation (or Summer Inter-Tropical Insolation Gradient, SITIG, Reichert et al., 1997) is critical, reflecting both “pull” and “push” forcings from the Northern and Southern Hemisphere that both propel monsoon changes (Cheng et al., 2022). Thus, we use the interhemispheric differential in tropical insolation (30°N-30°S) as an integrated insolation forcing of the monsoon, consistent with previous evidence, such as the Asian summer monsoon variation pattern over the MIS 3 (Cheng et al., 2022).*” In the manuscript, when referring previous studies, we keep the original description of insolation forcing (i.e., NHSI in most cases). However, for this study, we use the insolation differential (SITIG) as the insolation forcing in both text and figures.

6) Line 294-305: I find these lines repetitive. For brevity and for the benefit of the reader, the authors must remove these lines as they have already been discussed in the introduction.

Done.

7) Line 678: Are the changes in Precip over the Qunf cave significant?

The EC-Earth precipitation difference between 8K and PI at Qunf Cave is ~0.1mm/day. This may not be very large or significant, but regarding the P-E changes, model results show the same direction as those inferred from the multi-proxy records.

8) Supplementary material Line 101-103: The authors mention that precipitation is notably higher

at 8ka over the region of the cave. However, in Fig. 3b, it appears to have only marginally higher precipitation.

The precipitation at Qunf Cave at 8K was slightly higher than PI, the modelling result shows a ~ 0.1 mm/day decrease from 8K to PI.

Following the comment, we have changed the sentence in supplementary text (lines 101-102) to: “*The summer precipitation amount at Qunf Cave was marginally higher at 8K*”.

9) Supplementary fig 8: The authors have used four years to make a climatology plot. They should use at least 30 yrs, as done in supplementary figure 9.

Very good suggestion. We have replotted this figure using the ERA5 climatology data for 30 years (1981 - 2010), as same as supplementary Figure 9.

Supplementary Figure 8. 850 hPa wind of (a) pre-monsoon season (March to May, MAM), and (b) monsoon season (June to July) during 1981-2010 from ERA5.

D) Miscellaneous:

1) Line 23: Replace “climate meaning” with a more appropriate word.

We have replaced it with “*variability*”.

2) Line 25-26: It is not Indian summer monsoon moisture that prevails over southern Oman. In this context, it is better to say moisture from the Arabian Sea prevails in southern Oman.

We have adjusted this sentence to “*the moisture from the Arabian Sea prevails in southern Oman*”.

3. Line 164: Fig 3b is referred to first. Perhaps, it is better if the author's interchange figures 3a and 3b for maintaining flow.

Done.

4. Line 322: “between from”

We have deleted the word “between”. (Line 314)

5. Line 323: “a lower P-E. but”

We have corrected the sentence in line 315 to: “……a lower P-E. This requires further investigation.”

Reference

- Beck, J.W., Zhou, W.J., Li, C., Wu, Z.K., White, L., Xian, F., Kong, X.H., An, Z., 2018. A 550,000-year record of East Asian monsoon rainfall from Be-10 in loess. *Science* 360, 877-+.
- Cai, Y., An, Z., Cheng, H., Edwards, R.L., Kelly, M.J., Liu, W., Wang, X., Shen, C.-C., 2006. High-resolution absolute-dated Indian Monsoon record between 53 and 36 ka from Xiaobailong Cave, southwestern China. *Geology* 34, 621-624.
- Cheng, H., Li, H., Sha, L., Sinha, A., Shi, Z., Yin, Q., Lu, Z., Zhao, D., Cai, Y., Hu, Y., Hao, Q., Tian, J., Kathayat, G., Dong, X., Zhao, J., Zhang, H., 2022. Milankovitch theory and monsoon. *The Innovation*, 100338.
- Rohling, E.J., Abu-Zied, R., Casford, J., Hayes, A., Hoogakker, B., 2009. The marine environment: present and past. *The physical geography of the Mediterranean*, 33-67.

Reviewer #2

The authors have comprehensively responded to two detailed reviews, and I am satisfied with the authors' modifications in response to my comments (reviewer 2). I can now recommend this manuscript for publication following a few minor comments regarding the manuscript figures.

We are glad that you are satisfied with our response. Here are our responses to your following suggestions.

1. The top blue/yellow banner in Figure 2 needs to be described in the figure caption. It would also be helpful for the Holocene Humid Period grey bar to be identified in the figure caption.

We revised the color of Figure 2 and the caption as well:

Figure 2. Q5 multi-proxy records. (a) Q5 δD_{fi} , error bars are the standard deviation (1SD) of the reproducibility of crushed speleothem samples; (b) Q5 $\Delta^{17}O$; (c) Relative humidity calculated by Q5 $\Delta^{17}O$ data; (d) $\delta^{18}O_c$; (e) $\delta^{13}C$; (f) $(^{234}U/^{238}U)_0$; (g) Mg/Ca; (h) Sr/Ca; and (i) Ba/Ca. The top yellow and blue banners show the dominant monsoon system at the cave site, with ISM stands for Indian summer monsoon, and NASM stands for North African summer monsoon. The Holocene Humid Period is from 10.5 to 6 ka BP, coinciding with the intensified Northern African and Indian summer monsoons.

2) Please add the name of the model in the Figure 3 caption title, instead of just "Simulation results"

Good suggestion. We have adjusted the figure caption title to “**EC-Earth simulation results of summer (JJA) hydroclimate difference between 8K and PI.**”.

REVIEWERS' COMMENTS

Reviewer #1 (Remarks to the Author):

The authors have responded to my queries satisfactorily and I recommend publication of the manuscript. The authors have not answered my question regarding the EC-Earth simulations. There is no description of the simulations from EC-Earth that are used to calculate the RH values for time period between 0ka and 8ka. The authors have described only PI and 8ka simulations in the main text. Are these similar simulations with different boundary conditions? or are these RH values estimated from EC-Earth in some other way. Since, these simulations are not used in the main text, this does not affect my decision on the publication of the manuscript.

Point-by-point response

Reviewer 1#

The authors have responded to my queries satisfactorily and I recommend publication of the manuscript. The authors have not answered my question regarding the EC-Earth simulations. There is no description of the simulations from EC-Earth that are used to calculate the RH values for time period between 0ka and 8ka. The authors have described only PI and 8ka simulations in the main text. Are these similar simulations with different boundary conditions? or are these RH values estimated from EC-Earth in some other way. Since, these simulations are not used in the main text, this does not affect my decision on the publication of the manuscript.

We thank the reviewer for overall satisfaction with our responses.

The EC-Earth simulations we used to calculate the RH in the last response materials are the same as what we used in the main text, e.g. 8K and PI simulations. The calculated RH from 0 to 8ka, as shown in our last response, is from a Holocene transient simulation of EC-Earth3 that has not been published yet (only temperature data are used in Askjær et al., 2022), hence it is not used for the manuscript. The boundary conditions of EC-Earth 8K and PI simulations are shown in Supplementary Table 1.

Reference:

Askjær, T. G., Zhang, Q., Schenk, F., Ljungqvist, F. C., Lu, Z., Brierley, C. M., ... & Yang, H. (2022). Multi-centennial Holocene climate variability in proxy records and transient model simulations. *Quaternary Science Reviews*, 296, 107801.